# Breaking the Curse of Dimensionality for Parametric Elliptic PDEs

## Abstract

Motivated by recent empirical success, we examine how neural network-based ansatz classes can break the curse of dimensionality for high-dimensional, non-linear elliptic partial differential equations (PDEs) with variational structure. The high-dimensionality of the PDEs can either be induced through a high-dimensional physical domain or a high-dimensional parameter space. The latter include parametric right-hand sides, parametric domains, and material constants. Our main result shows that any scheme that computes neural network based $W^{1,p}$-approximations, leverages the extraordinary approximation capabilities of neural networks and, thus, is able to beat the curse of dimensionality if the ground truth solution is smooth or possesses Barron regularity. Popular examples of $W^{1,p}$-convergent schemes include, e.g., the Deep Ritz Method and physics-informed neural networks. We present numerical experiments supporting our theoretical findings.

## 1 Introduction

High-dimensional partial differential equations (PDEs) arise naturally in applications with either a high-dimensional domain, a high-dimensional parameter space, or possibly with both. The former includes the Schrödinger equation in quantum physics, the Black–Scholes equation in finance, and the Hamilton–Jacobi–Bellman equation in control theory, we refer to Weinan et al. (2021); Bellman (1954). On the other hand, examples of problems with high-dimensional parameter space are ubiquitous in engineering applications, for instance, in varying material properties, right-hand sides or even in the form of varying computational domains, as discussed in Hennigh et al. (2021); Ohlberger & Rave (2016).

For problems with a high-dimensional physical domain, classical mesh-based approximation schemes face the curse of dimensionality, meaning that the computational cost increases exponentially with the dimension of the problem. In the case of parametric problems, one is typically interested in querying the PDE solution for many different parameter instances, possibly with low inference time. To this end, classical methods need to repeatedly solve the equations for every required parameter instance, a potentially prohibitively expensive or slow computational task, see Biegler et al. (2007). Even assuming additional, favorable structure of the solution of a high-dimensional PDE – may it be a latent low-dimensionality of the solution or a high degree of smoothness – it remains a challenge for classical methods to approximate the solution with an acceptable accuracy, especially in situations of non-linear solution manifolds as discussed in Ohlberger & Rave (2016); Lee & Carlberg (2020).

Artificial neural networks have shown great potential in the approximation of high-dimensional functions, among those computer vision, classification and natural language processing tasks and are known to possess extraordinary approximation capabilities with the possibility to achieve dimension-independent approximation rates for certain function classes, see Ma et al. (2022); Barron (1993); Yarotsky (2017); Gühring & Raslan (2021); Gühring et al. (2020). Therefore, investigating artificial neural networks as ansatz classes for the solution of PDEs or PDE solution operators has recently gained increased interest for high-dimensional and parametric problems. We refer to Kutyniok et al. (2022); Weinan & Wojtowytsch (2022); Jentzen et al. (2021); Chen et al. (2021) for theoretical studies. Successful empirical results of neural network-based applications to PDEs posed in high-dimensional spaces include Hermann et al. (2020); Yu & E (2018); Han et al. (2018); Sirignano & Spiliopoulos (2018). For the parametric setting, we direct the reader to Li et al. (2021); Khoo et al. (2021); Lee & Carlberg (2020); Geist et al. (2021).

Of the aforementioned contributions, the theoretical works either focus on approximation theoretic results or consider linear problems without parametric dependencies. We discuss the relation to our contribution in detail in Section 1.1. The approximation theoretic results guarantee the existence of a neural network with desirable approximation rates but provide no practical way to compute the neural network. For non-linear, parametric PDEs with $p$-structure, our results deliver the necessary PDE analysis – in a setting suitable for neural network ansatz functions – to alleviate this problem. Instead of explicitly constructing a neural network, we show that it suffices to find a neural network approximation that is close "in energy" to the ground truth solution. This makes a significant difference: Energy approximations can be found by using the variational energy as a loss function or more generally by any $W^{1,p}$-convergent approximation scheme – a natural property of a reasonable approximation algorithm.

To summarize, our main contributions are the following:

- We show that every energy convergent approximation scheme for the $p$-Dirichlet energy utilizing neural network ansatz functions can leverage the extraordinary approximation capabilities of neural networks. Further, we explain which assumptions on the ground truth solution allow neural networks to beat the curse of dimensionality. We extend these results to parametric problems, where the neural network approximates simultaneously in the physical and the parameter space. Our contributions are the first quantitative error estimates for parametric and non-linear elliptic PDEs for neural network approximation schemes.
- From a mathematical point of view, the analysis of non-linear ansatz classes is novel in the case of the $p$-Laplacian. Existing literature exclusively exploits strategies based on optimality conditions (Galerkin orthogonality) only available for linear ansatz classes, hence, excludes neural networks. Further, to the best of the author's knowledge, error estimates for parametric problems have not been considered in the existing literature.

## 1.1 Main Result and Related Work

For clarity, we present our main result for the case of a parametric right-hand side and with homogeneous Neumann boundary conditions. However, different boundary conditions and parametric dependencies are covered by our analysis. We explain this in the Appendix and refer to Section C.

Consider a physical domain $\Omega \subseteq \mathbb{R}^{d_\Omega}$, $d_\Omega \in \mathbb{N}$, and a parameter space $\mathcal{P} \subseteq \mathbb{R}^{d_\mathcal{P}}$, $d_\mathcal{P} \in \mathbb{N}$. Further, let $p \in (1, \infty)$ be fixed and denote by $(\boldsymbol{f}(\boldsymbol{\tau}, \cdot))_{\boldsymbol{\tau} \in \mathcal{P}}$ a parametric family of right-hand sides. We study the non-linear $p$-Laplace problem as a prototypical, non-linear elliptic PDE. More precisely, we want to find $\boldsymbol{u}^* : \mathcal{P} \times \Omega \to \mathbb{R}$ satisfying

$$-\operatorname{div}\left(|\nabla_x \boldsymbol{u}^*(\boldsymbol{\tau}, x)|^{p-2} \nabla_x \boldsymbol{u}^*(\boldsymbol{\tau}, x)\right) = \boldsymbol{f}(\boldsymbol{\tau}, x) \quad \text{for a.e. } (\boldsymbol{\tau}, x)^\top \in \mathcal{P} \times \Omega, \tag{1}$$

subjected to – for simplicity – homogeneous Neumann boundary conditions. The case $p = 2$ retrieves the classical Poisson equation. In this example, the parametric dependencies are induced through the right-hand side and both $\Omega$ and $\mathcal{P}$ may be high-dimensional. Then, we seek a neural network $\boldsymbol{u}_\theta$ with input $(\boldsymbol{\tau}, x)^\top \in \mathbb{R}^{d_\mathcal{P}} \times \mathbb{R}^{d_\Omega}$ that approximates the solution $\boldsymbol{u}^* : \mathcal{P} \times \Omega \to \mathbb{R}$ simultaneously in the physical domain $\Omega$ and the parameter space $\mathcal{P}$. Essential for the statement of our result is the reformulation of equation equation 1 as a minimization problem. We find $\boldsymbol{u}^* \in L^p(\mathcal{P}, W^{1,p}(\Omega))$ as a minimizer of $\boldsymbol{\mathcal{E}} : L^p(\mathcal{P}, W^{1,p}(\Omega)) \to \mathbb{R}$, for every $\boldsymbol{v} \in L^p(\mathcal{P}, W^{1,p}(\Omega))$ defined by

$$\boldsymbol{\mathcal{E}}(\boldsymbol{v}) := \int_\mathcal{P} \left[ \frac{1}{p} \int_\Omega |\nabla_x \boldsymbol{v}(\boldsymbol{\tau}, x)|^p \, \mathrm{d}x - \int_\Omega \boldsymbol{f}(\boldsymbol{\tau}, x) \, \boldsymbol{v}(\boldsymbol{\tau}, x) \, \mathrm{d}x \right] \mathrm{d}\boldsymbol{\tau} \,. \tag{2}$$

Then, our main result is the following.

**Theorem 1.** *Let $\Omega \subseteq \mathbb{R}^{d_\Omega}$, $d_\Omega \in \mathbb{N}$, be a bounded Lipschitz domain and $\mathcal{P} \subseteq \mathbb{R}^{d_\mathcal{P}}$, $d_\mathcal{P} \in \mathbb{N}$, an open set. Moreover, let $\boldsymbol{f} \in L^{p'}(\mathcal{P} \times \Omega)$, $p \in (1, \infty)$, be such that $\int_\Omega \boldsymbol{f}(\boldsymbol{\tau}, \cdot) \, \mathrm{d}x = 0$ for a.e. $\boldsymbol{\tau} \in \mathcal{P}$. Denote by $\boldsymbol{u}^* \in L^p(\mathcal{P}, W^{1,p}(\Omega))$, a weak solution of the parametric $p$-Laplace problem with homogeneous Neumann boundary conditions, i.e.,*

$$\begin{aligned} -\operatorname{div}(|\nabla_x \boldsymbol{u}^*|^{p-2} \nabla_x \boldsymbol{u}^*) &= \boldsymbol{f} && \text{in } \mathcal{P} \times \Omega \,, \\ \partial_n \boldsymbol{u}^* &= 0 && \text{on } \mathcal{P} \times \partial\Omega \,. \end{aligned} \tag{3}$$

*Let $M \subset W^{1,p}(\mathcal{P} \times \Omega)$ be any subset that contains the zero function[1] and let $\boldsymbol{v} \in M$ be arbitrary. Setting*

$$\tilde{M} := \left\{ \boldsymbol{u} \in M \mid \|\nabla_x \boldsymbol{u}\|_{L^p(\mathcal{P} \times \Omega)^d} \leq 2\|\nabla_x \boldsymbol{u}^*\|_{L^p(\mathcal{P} \times \Omega)^d} \right\},$$

*it holds*

$$\|\nabla_x \boldsymbol{v} - \nabla_x \boldsymbol{u}^*\|_{L^p(\mathcal{P} \times \Omega)^d} \precsim \begin{cases} \delta(\boldsymbol{v})^{1/p} + \inf_{\tilde{\boldsymbol{v}} \in \tilde{M}} \|\nabla_x \tilde{\boldsymbol{v}} - \nabla_x \boldsymbol{u}^*\|_{L^p(\mathcal{P} \times \Omega)^d}^{\frac{2}{p}} & \text{if } p \in [2, \infty) \\ \delta(\boldsymbol{v})^{1/2} + \inf_{\tilde{\boldsymbol{v}} \in \tilde{M}} \|\nabla_x \tilde{\boldsymbol{v}} - \nabla_x \boldsymbol{u}^*\|_{L^p(\mathcal{P} \times \Omega)^d}^{\frac{p}{2}} & \text{if } p \in (1, 2) \end{cases},$$

*where $\delta(\boldsymbol{v}) := \mathcal{E}(\boldsymbol{v}) - \inf_{\tilde{\boldsymbol{v}} \in \tilde{M}} \mathcal{E}(\tilde{\boldsymbol{v}})$ is the optimization error and the implicit constants depend on $p$, $\Omega$ and $\|\boldsymbol{f}\|_{L^{p'}(\mathcal{P} \times \Omega)}$ only.*

We stress again that the choice of Neumann boundary conditions is for simplicity of presentation. A similar result holds for Dirichlet boundary conditions employing an appropriate penalty scheme. The fact that $\boldsymbol{f}(\boldsymbol{\tau}, \cdot)$ for a.e. $\boldsymbol{\tau} \in \mathcal{P}$ is mean-value-free serves to guarantee the well-posedness in the Neumann boundary value case. The main reason to pass from $M$ to $\tilde{M}$ is solely of technical nature, rooted in Lemma 7 and not of relevance for the interpretation of the result.

**Leveraging the Power of Approximation-Theoretical Results.** As we discuss below, suitable approximation theorems allow to estimate the infimum in Theorem 1 to deduce error decay rates. Note that there is no further requirement on the approximating function $\boldsymbol{v} \in M$ than that it is a "good" quasi-optimizer of $\mathcal{E} : M \to \mathbb{R}$, i.e., that $\delta(\boldsymbol{v})$ is sufficiently small. Thus, any algorithm that produces approximate solutions that converge in energy is able to fully leverage the approximation capabilities of neural network ansatz classes – up to the exponent $2/p$ or $p/2$, which is due to the non-linearity. Furthermore, energy convergence is equivalent to convergence in the $W^{1,p}$-semi-norm[2] We stress the drastic difference of our contribution to mere approximation theoretical results that only guarantee the existence of a well-approximating network, yet don't unveil how such an approximation should be found. In this sense, our contribution is orthogonal to approximation theoretical results as it can be combined with these to extend them.

Finally note that, analyzing the effect of a solution scheme on the achievable value of $\delta(\boldsymbol{v})$ is a difficult problem, typically connected to a non-convex optimization task, that we do not study in this article.

**Using Smoothness to Beat the Curse of Dimensionality.** We can utilize quantitative universal approximation results to estimate the infimum in Theorem 1. In some situations, this allows us to beat the curse of dimensionality. Assume that the solution $\boldsymbol{u}^*$ to equation 3 is a member of $W^{k,p}(\mathcal{P} \times \Omega)$ for some $k \in \mathbb{N}$, $k > 1$. Then, for every $n \in \mathbb{N}$, we may use Theorem 4.9 in Gühring & Raslan (2021) to guarantee the existence of a fully connected neural network architecture with ReLU[2]-activation[3] with parameter space $\Theta_n$ of dimension $\mathcal{O}(n)$ such that, setting $M = \mathcal{F}_{\Theta_n}$, where $\mathcal{F}_{\Theta_n}$ denotes the realization set of the ansatz class, it holds

$$\inf_{\psi \in \Theta_n} \|\nabla_x \boldsymbol{u}_\psi - \nabla_x \boldsymbol{u}^*\|_{L^p(\mathcal{P} \times \Omega)^d} \precsim \|\boldsymbol{u}^*\|_{W^{k,p}(\mathcal{P} \times \Omega)}^{\frac{2}{p}} \left(\frac{1}{n}\right)^{\frac{2}{p} \frac{k-1}{d_\Omega + d_\mathcal{P}}}$$

for the case $p \geq 2$ and with $\frac{2}{p}$ replaced by $\frac{p}{2}$ in the case $p < 2$. Hence, for arbitrary $\boldsymbol{u}_\theta \in M$, $\theta \in \Theta_n$, we get

$$\|\nabla_x \boldsymbol{u}_\theta - \nabla_x \boldsymbol{u}^*\|_{L^p(\mathcal{P} \times \Omega)^d} \precsim \begin{cases} \delta_n(\boldsymbol{u}_\theta)^{1/p} + \|\boldsymbol{u}^*\|_{W^{k,p}(\mathcal{P} \times \Omega)} \left(\frac{1}{n}\right)^{\frac{2}{p} \cdot \frac{k-1}{d_\Omega + d_\mathcal{P}}} & \text{if } p \in [2, \infty) \\ \delta_n(\boldsymbol{u}_\theta)^{1/2} + \|\boldsymbol{u}^*\|_{W^{k,p}(\mathcal{P} \times \Omega)} \left(\frac{1}{n}\right)^{\frac{p}{2} \cdot \frac{k-1}{d_\Omega + d_\mathcal{P}}} & \text{if } p \in (1, 2) \end{cases},$$

where $\delta_n(\boldsymbol{u}_\theta) := \mathcal{E}(\boldsymbol{u}_\theta) - \inf_{\psi \in \Theta_n} \mathcal{E}(\boldsymbol{u}_\psi)$. This shows – given sufficient smoothness of $\boldsymbol{u}^* \in M$ – that the error of the neural network approximation does not decay exponentially slow in the dimension $n \in \mathbb{N}$ of the parameter space $\Theta_n$. More precisely, the result requires dimension-dependent smoothness with the smoothness parameter $k \in \mathbb{N}$, $k > 1$, scaling like $k \sim d_\Omega + d_\mathcal{P}$. However, the assumption of smoothness is very natural in the context of (linear) elliptic PDEs and holds also in the parametric case, see Lemma B which gives an easily verifiable criterion when the smoothness assumption holds.

---

[1]The subsets we have in mind consist of neural network functions of a given architecture and, thus, $\boldsymbol{v} \in M$ is a neural network. But any choice of $M$ is admissible.

[2]To see this, we refer to equation 7 in Proposition 2.

[3]This result holds also for other activation functions, we refer to the original work.

**Employing Barron Regularity to Beat the Curse of Dimensionality.** A different situation where the curse of dimensionality can be circumvented is when $\boldsymbol{u}^* \in M$ is a member of the Barron space $\mathcal{B}$, or can be "well-approximated" by Barron functions. We refer to Barron (1993); Ma et al. (2022) for a definition of the Barron space. In essence, members of $\mathcal{B}$ can be approximated with respect to the $H^1$-norm by shallow neural networks with a dimension-independent rate of $n^{-1/2}$, where $n \in \mathbb{N}$ is the width of the shallow network. Hence, setting $p = 2$ and assuming $\boldsymbol{u}^* \in \mathcal{B}$, we can estimate for an arbitrary shallow neural network $\boldsymbol{u}_\theta \in M$

$$\|\nabla_x \boldsymbol{u}_\theta - \nabla_x \boldsymbol{u}^*\|_{L^2(\mathcal{P} \times \Omega)^d} \lesssim \delta_n^{\frac{1}{2}} + \left(\frac{1}{n}\right)^{\frac{1}{2}} \|\boldsymbol{u}^*\|_{\mathcal{B}}.$$

The assumption $\boldsymbol{u}^* \in \mathcal{B}$ is too restrictive in general, cf. the discussion in Weinan & Wojtowytsch (2022). However, assuming that the data $\boldsymbol{f} \in L^2(\mathcal{P} \times \Omega)$ (and possibly coefficients) are of Barron regularity, it was recently established that the solution $\boldsymbol{u}^* \in M$ can be approximated by Barron functions with Barron norm growing only polynomially in the dimension, yielding the rate $n^{-1/2}$ for shallow networks of width $(dn)^{C \log(n)}$, where $C$ is a constant, we refer to Chen et al. (2021). Note that the result in Chen et al. (2021) so far does only hold for linear elliptic PDEs and special activation functions, and does not include parametric dependencies. Our result, then, shows that this error decay rate is preserved for any energy-convergent approximation scheme and is, in fact, not a mere approximation result.

**Related Work** For PDEs that admit a stochastic representation, several situation are known in which the curse of dimensionality can be circumvented, Jentzen et al. (2021); Han et al. (2018); Weinan et al. (2021). These results are of approximation theoretic nature and do not provide a way to construct the approximating network. Further, this approach crucially relies on the stochastic representation of the PDE's solution and, thus, is not generally applicable.

The works Xu (2020); Jiao et al. (2021); Duan et al. (2021); Müller & Zeinhofer (2022) are similar to our contribution since they consider elliptic equations and provide a Céa type Lemma and, consequently, are not only approximation theoretic results. However, they neither analyze non-linear nor parametric equations.

The contributions in Chen et al. (2021); Weinan & Wojtowytsch (2022) mark the beginning of a regularity theory for elliptic equations with respect to Barron spaces. These results are complementary to our analysis in the sense that they can be combined with our contribution. For instance, the main result of Chen et al. (2021) states that a solution to a linear elliptic PDE with Barron data is "almost" of Barron regularity and can be approximated with a polynomial rate with respect to the dimension. Our analysis, then, guarantees that every neural network approximation that is close in energy to the ground truth solution realizes this rate.

**Notation** For a Banach space $X$ with norm $\|\cdot\|_X : X \to \mathbb{R}_{\geq 0}$, we denote by $X^*$, its (topological) dual space equipped with the norm $\|\cdot\|_{X^*} : X^* \to \mathbb{R}_{\geq 0}$, defined by $\|x^*\|_{X^*} := \sup_{\|x\|_X \leq 1} \langle x^*, x \rangle_X$ for all $x^* \in X^*$. Here, $\langle \cdot, \cdot \rangle_X : X^* \times X \to \mathbb{R}$ denotes the duality pairing, defined by $\langle x^*, x \rangle_X := x^*(x)$ for all $x^* \in X^*$ and $x \in X$.

For $p \in [1, \infty]$, we denote by $L^p(\Omega)$, the space of (Lebesge-)measurable functions $u : \Omega \to \mathbb{R}$ that are integrable in $p$-th power, i.e., $\int_\Omega |u|^p \, \mathrm{d}x < \infty$ if $p \in [1, \infty)$ and $\mathrm{ess\,sup}_{x \in \Omega} |u(x)| < \infty$ if $p = \infty$. Endowed with the norm $\|u\|_{L^p(\Omega)} := (\int_\Omega |u|^p \, \mathrm{d}x)^{\frac{1}{p}}$ if $p \in [1, \infty)$ and $\|u\|_{L^\infty(\Omega)} := \mathrm{ess\,sup}_{x \in \Omega} |u(x)|$ if $p = \infty$, the space $L^p(\Omega)$ forms a Banach space, which is separable if $p \in [1, \infty)$ and reflexive if $p \in (1, \infty)$, cf. (Adams & Fournier, 2003, Chapter 2).

For $k \in \mathbb{N}$ and $p \in [1, \infty]$, we denote by $W^{k,p}(\Omega)$, the space of functions in $L^p(\Omega)$ with distributional derivatives up to $k$-th order in $L^p(\Omega)$. Endowed with the norm $\|u\|_{W^{k,p}(\Omega)} := \sum_{l=0}^k \|D^l u\|_{L^p(\Omega)}$, the space $W^{k,p}(\Omega)$ forms a Banach space, which is separable if $p \in [1, \infty)$ and reflexive if $p \in (1, \infty)$, cf. (Adams & Fournier, 2003, Chapter 3). For $k \in \mathbb{N}$ and $p \in [1, \infty]$, we denote by $W_0^{k,p}(\Omega)$, the closure of all compactly supported, smooth functions $C_c^\infty(\Omega)$ in $W^{k,p}(\Omega)$.

We always denote parameter-dependent functions by boldface letters, e.g., $\boldsymbol{u}, \boldsymbol{v}, \boldsymbol{w}, \ldots$, and parameter-independent functions by non-boldface letters, e.g., $u, v, w, \ldots$. In the same spirit, we denote by $\mathcal{E} : L^p(\mathcal{P}; W^{1,p}(\Omega)) \to \mathbb{R}$, the parametric $p$-Dirichlet energy equation 2 and by $E : W^{1,p}(\Omega) \to \mathbb{R}$, the non-parametric $p$-Dirichlet energy (c.f. equation 6).

## 2 PROOF OF THE MAIN RESULT

In this section, we provide the proof of Theorem 1. For clarity, we first consider the non-parametric case and extend the results afterwards to include parametric dependencies.

### 2.1 PROOF OF THE NON-PARAMETRIC SETTING

The main step in the proof of the non-parametric version of Theorem 1 is to show that convergence in energy, i.e., $E(u_n) \to E(u^*)$ $(n \to \infty)$ (cf. equation 6) for a neural network approximation $u_n \in M$ of the ground truth solution $u^* \in M$, is equivalent to the convergence of $u_n \to u^*$ $(n \to \infty)$ in the Sobolev topology. To establish this in a quantitative fashion, we need an optimal measure of the convexity of the $p$-Dirichlet energy (cf. equation 6). This is given through the bi-variate, symmetric mapping $\rho_F^2 : W^{1,p}(\Omega) \times W^{1,p}(\Omega) \to \mathbb{R}$, defined by

$$\rho_F^2(v,w) := \|F(\nabla v) - F(\nabla w)\|_{L^2(\Omega)^d}^2 \quad \text{for all } v, w \in W^{1,p}(\Omega), \tag{4}$$

where $F : \mathbb{R}^d \to \mathbb{R}^d$ is defined by

$$F(a) := |a|^{\frac{p-2}{2}} a \quad \text{for all } a \in \mathbb{R}^d. \tag{5}$$

The map $\rho_F^2 : W^{1,p}(\Omega) \times W^{1,p}(\Omega) \to \mathbb{R}$ is the optimal distance measure for the $p$-Dirichlet problem. This is embodied in the two-sided estimate proved in the next proposition, see equation 7, that relates convergence in energy to convergence in terms of $\rho_F^2 : W^{1,p}(\Omega) \times W^{1,p}(\Omega) \to \mathbb{R}$. In the literature, $\rho_F^2 : W^{1,p}(\Omega) \times W^{1,p}(\Omega) \to \mathbb{R}$ is usually referred to as the *Natural Distance*, cf. Diening & Růžička (2007); Diening et al. (2007); Diening & Ettwein (2008); Kaltenbach & Růžička (2022).

Next, we establish a Céa type Lemma in terms of $\rho_F^2 : W^{1,p}(\Omega) \times W^{1,p}(\Omega) \to \mathbb{R}$, see Lemma 5. This decomposes the distance of $u_n$ to $u^*$ into an optimization error and an approximation theoretic contribution. Finally, we study the relation of $\rho_F^2 : W^{1,p}(\Omega) \times W^{1,p}(\Omega) \to \mathbb{R}$ to the standard Sobolev topology in Lemma 7.

From a technical perspective, the central estimate in equation 7 is proved via a Taylor expansion of the $p$-Dirichlet energy around its minimizer $u^* \in W^{1,p}(\Omega)$. However, care needs to be taken since $E : W^{1,p}(\Omega) \to \mathbb{R}$ is not twice continuously differentiable and a subtle regularization procedure needs to be employed to rigorously carry out the expansion.

**Proposition 2.** *Let $\Omega \subseteq \mathbb{R}^d$, $d \in \mathbb{N}$, be a bounded domain, $f \in W^{1,p}(\Omega)^*$, $p \in (1,\infty)$, and let $U \subseteq W^{1,p}(\Omega)$ be a closed subspace such that Poincaré's inequality is valid, i.e., there exists a constant $C_P > 0$ such that for every $v \in U$, it holds*

$$\|v\|_{L^p(\Omega)} \le C_P \|\nabla v\|_{L^p(\Omega)^d}.$$

*Moreover, define $E : U \to \mathbb{R}$ for every $v \in U$ by*

$$E(v) := \frac{1}{p} \int_\Omega |\nabla v|^p \, \mathrm{d}x - \langle f, v \rangle_{W^{1,p}(\Omega)}. \tag{6}$$

*Then, the following statements apply:*

*(i) There exists a unique minimizer $u^* \in U$ for $E : U \to \mathbb{R}$.*

*(ii) There exists a constant $c(p) > 0$, depending only on $p \in (1,\infty)$ and not depending on $d \in \mathbb{N}$, such that for every $v \in U$, it holds*

$$c(p)^{-1} \rho_F^2(v, u^*) \le E(v) - E(u^*) \le c(p) \rho_F^2(v, u^*). \tag{7}$$

*Moreover, we can choose $c(p) > 0$ such that $(p \mapsto c(p)) \in C^0(1,\infty)$.*

*Proof.* The proof is provided in the Appendix, see A. $\qquad\square$

**Remark 3** (The Case $p = 2$)**.** In the case $p = 2$, we retrieve the well-known Dirichlet energy. Further, equality holds in equation 7 with constant $c(p) = \frac{1}{2}$. More precisely, for every $v \in U$, we have that

$$E(v) - E(u^*) = \frac{1}{2}\|\nabla v - \nabla u^*\|_{L^2(\Omega)^d}^2 = \rho_F^2(v, u^*).$$

This can be shown by a straight-forward Taylor expansion of $E : U \to \mathbb{R}$ around $u^* \in U$, cf. Müller & Zeinhofer (2022).

**Remark 4** (The Role of the Space $U$). The space $U$ encodes boundary conditions, for example, $U = W_0^{1,p}(\Omega)$ is an admissible choice. However, when choosing $U = W^{1,p}(\Omega)$ and requiring that the right-hand side $f \in W^{1,p}(\Omega)^*$ vanishes on constant functions, Proposition 2 stays valid with the exemption of the uniqueness of the minimizer $u^* \in U$. In this case, $u^* \in U$ is only determined up to additive constants. This can be seen by considering the energy $E$ on the quotient space $W^{1,p}(\Omega)$ modulo the constant functions. On this space, a Poincaré inequality is available.

An immediate consequence of Theorem 2 is the following Céa type lemma.

**Lemma 5** (Céa Lemma). *Let the assumptions of Proposition 2 be satisfied. Moreover, let $M \subseteq U$ be an arbitrary subset. Then, there exists a constant $c(p) > 0$, depending only on $p \in (1, \infty)$ and not depending on $d \in \mathbb{N}$, such that for every $v \in M$, it holds*

$$\rho_F^2(v, u^*) \le c(p) \left( \delta(v) + \inf_{\tilde{v} \in M} \rho_F^2(\tilde{v}, u^*) \right),$$

*where $\delta(v) := E(v) - \inf_{\tilde{v} \in M} E(\tilde{v})$ is the optimization error. Moreover, we can choose $c(p) > 0$ such that $(p \mapsto c(p)) \in C^0(1, \infty)$.*

*Proof of Lemma 5.* Let $v \in M$ be arbitrary. Then, by referring to Theorem 2, we find that

$$c(p)^{-1} \rho_F^2(v, u^*) \le E(v) - \inf_{\tilde{v} \in M} E(\tilde{v}) + \inf_{\tilde{v} \in M} E(\tilde{v}) - E(u^*) \le \delta(v) + c(p) \inf_{\tilde{v} \in M} \rho_F^2(\tilde{v}, u). \quad \square$$

**Remark 6.** Note that we do not need to impose any structure on the set $M$, in particular, it does not need to possess a linear structure. This, in contrast to classical formulations of Céa's Lemma, allows us to choose $M$ as an ansatz class consisting of neural networks.

In order to arrive at error decay rates in Sobolev topology, we need the relation of $\rho_F^2$ to the $W^{1,p}(\Omega)$-semi-norm.

**Lemma 7** (Relation Between Natural Distance and $W^{1,p}$-Semi-Norm). *Let $\Omega \subseteq \mathbb{R}^d$, $d \in \mathbb{N}$, be a bounded domain and $p \in (1, \infty)$. Then, there exists a constant $c(p) > 0$, depending only on $p \in (1, \infty)$ and not depending on $d \in \mathbb{N}$, such that the following relations apply:*

*(i) If $p \in [2, \infty)$, then for every $u, v \in W^{1,p}(\Omega)$, it holds*

$$c(p)^{-1} \|\nabla u - \nabla v\|_{L^p(\Omega)^d}^p \le \rho_F^2(u, v)$$
$$\le c(p) \left( \|\nabla u\|_{L^p(\Omega)^d} + \|\nabla v\|_{L^p(\Omega)^d} \right)^{p-2} \|\nabla u - \nabla v\|_{L^p(\Omega)^d}^2.$$

*(ii) If $p \in (1, 2)$, then for every $v, w \in W^{1,p}(\Omega)$, it holds*

$$c(p)^{-1} \rho_F^2(u, v) \le \|\nabla u - \nabla v\|_{L^p(\Omega)^d}^p$$
$$\le c(p) \left( \|\nabla u\|_{L^p(\Omega)^d} + \|\nabla v\|_{L^p(\Omega)^d} \right)^{\frac{p(2-p)}{2}} \rho_F^2(u, v)^{\frac{p}{2}}.$$

*Moreover, we can choose $c(p) > 0$ such that $(p \mapsto c(p)) \in C^0(1, \infty)$.*

*Proof.* The proof is provided in the Appendix, see A. $\quad \square$

We are now able to prove the main result in a setting excluding parametric dependencies.

**Theorem 8.** *Let $f \in W^{1,p}(\Omega)^*$, $p \in (1, \infty)$, be such that $\langle f, c \rangle_{W^{1,p}(\Omega)} = 0$ for all $c \in \mathbb{R}$. Moreover, let $u^* \in W^{1,p}(\Omega)$ be a weak solution of the $p$-Laplace problem with homogeneous Neumann boundary conditions, i.e., $u^* \in W^{1,p}(\Omega)$ minimizes $E : W^{1,p}(\Omega) \to \mathbb{R}$, for every $v \in W^{1,p}(\Omega)$ defined by*

$$E(v) := \frac{1}{p} \int_\Omega |\nabla v|^p \, dx - \langle f, v \rangle_{W^{1,p}(\Omega)}. \tag{8}$$

*Let $M \subset W^{1,p}(\Omega)$ be any subset that contains the zero function and let $v \in M$ be an arbitrary. Setting*

$$\tilde{M} := \left\{ u \in M \mid \|\nabla u\|_{L^p(\Omega)^d} \le 2 \|\nabla u^*\|_{L^p(\Omega)^d} \right\},$$

*it holds*

$$\|\nabla v - \nabla u^*\|_{L^p(\Omega)^d} \lesssim \begin{cases} \delta(v)^{1/p} + \inf_{\tilde{v} \in \tilde{M}} \|\nabla \tilde{v} - \nabla u^*\|_{L^p(\Omega)^d}^{\frac{2}{p}} & \text{if } p \in [2, \infty) \\ \delta(v)^{1/2} + \inf_{\tilde{v} \in \tilde{M}} \|\nabla \tilde{v} - \nabla u^*\|_{L^p(\Omega)^d}^{\frac{p}{2}} & \text{if } p \in (1, 2) \end{cases},$$

*where $\delta(v) := E(v) - \inf_{\tilde{v} \in \tilde{M}} E(\tilde{v})$ is the optimization error and the implicit constant depends on $p$, $\Omega$ and $\|f\|_{W^{1,p}(\Omega)^*}$ only.*

*Proof. ad $p \in [2, \infty)$.* If $p \in [2, \infty)$, then we estimate, using the relation of the natural distances to Sobolev semi-norms as described in Lemma 7, Céa's Lemma 5, and the coercivity estimate in Lemma 13 to obtain

$$\|\nabla v - \nabla u^*\|_{L^p(\Omega)^d}^p \leq c(p)\, \rho_F^2(v, u^*)$$

$$\leq c(p) \left( \delta(v) + \inf_{\tilde{v}} \rho_F^2(\tilde{v}, u^*) \right)$$

$$\leq c(p) \left( \delta(v) + \inf_{\tilde{v} \in \tilde{M}} \left[ \left( \|\nabla \tilde{v}\|_{L^p(\Omega)^d} + \|\nabla u^*\|_{L^p(\Omega)^d} \right)^{p-2} \|\nabla \tilde{v} - \nabla u^*\|_{L^p(\Omega)^d}^2 \right] \right)$$

$$\leq c(p) \left( \delta(v) + 3^{p-2} \|\nabla u^*\|_{L^p(\Omega)^d}^{p-2} \inf_{\tilde{v} \in \tilde{M}} \|\nabla \tilde{v} - \nabla u^*\|_{L^p(\Omega)^d}^2 \right)$$

$$\leq c(p)\, \delta(v) + 3^{p-2}\, c(p, \Omega) \|f\|_{W^{1,p}(\Omega)^*}^{\frac{p-2}{p-1}} \inf_{\tilde{v} \in \tilde{M}} \|\nabla \tilde{v} - \nabla u^*\|_{L^p(\Omega)^d}^2.$$

*ad $p \in (1, 2]$.* If $p \in (1, 2]$, then, again, using the relation of the natural distance to Sobolev semi-norms (cf. Lemma 7) and Céa's Lemma 5, we obtain

$$\|\nabla v - \nabla u^*\|_{L^p(\Omega)^d} \leq c(p) \left( \|\nabla v\|_{L^p(\Omega)^d} + \|\nabla u^*\|_{L^p(\Omega)^d} \right)^{\frac{2-p}{2}} \left( \delta(v)^{\frac{1}{2}} + \inf_{\tilde{v} \in M} \|\nabla \tilde{v} - \nabla u^*\|_{L^p(\Omega)^d}^{\frac{p}{2}} \right).$$

Thus, it remains to estimate the first factor in the equation above. We use the coercivity estimate of Lemma 13 to obtain

$$\|\nabla v\|_{L^p(\Omega)^d} \leq c(p, \Omega) \left( E(v) + \|f\|_{W^{1,p}(\Omega)^*}^{p'} \right)^{\frac{1}{p}} \leq c(p, \Omega) \|f\|_{W^{1,p}(\Omega)^*}^{\frac{1}{p-1}},$$

where we used that $0 \in M$ to be able to estimate $E(v) \leq \delta(v)$. As a result, again applying Lemma 13, it follows that

$$\left( \|\nabla v\|_{L^p(\Omega)^d} + \|\nabla u^*\|_{L^p(\Omega)^d} \right)^{\frac{2-p}{2}} \leq c(p, \Omega) \left( \delta(v)^{\frac{2-p}{2p}} + \|f\|_{W^{1,p}(\Omega)^*}^{\frac{2-p}{2p-2}} \right)$$

$$= c(p, \Omega) \left( \delta(v)^{\frac{1}{2}} + \|f\|_{W^{1,p}(\Omega)^*}^{\frac{2-p}{2p-2}} \right).$$

Assuming $\delta(v) \leq 1$, it holds $\delta(v)^{\frac{1}{2}} + \delta(v)^{\frac{1}{p}} \leq 2\delta(v)^{\frac{1}{2}}$, which concludes the proof. $\square$

## 2.2 Proof of the Parametric Setting

As detailed in the introduction, the energy formulation we use for a $p$-Laplace problem with a parametric right-hand side $\boldsymbol{f} \in L^{p'}(\mathcal{P} \times \Omega)$ and parameter space $\mathcal{P} \subseteq \mathbb{R}^{d_\mathcal{P}}$, $d_\mathcal{P} \in \mathbb{N}$, for every $\boldsymbol{v} \in L^p(\mathcal{P}, W^{1,p}(\Omega))$, is defined by

$$\boldsymbol{\mathcal{E}}(\boldsymbol{v}) := \int_\mathcal{P} \left[ \frac{1}{p} \int_\Omega |\nabla_x \boldsymbol{v}(\boldsymbol{\tau}, \cdot)|^p \, dx - \int_\Omega \boldsymbol{f}(\boldsymbol{\tau}, \cdot)\, \boldsymbol{v}(\boldsymbol{\tau}, \cdot) \, dx \right] d\boldsymbol{\tau}.$$

Before proving the error decay rates of Theorem 1, we need to identify the correct function space $\mathcal{U}$ for the definition of $\boldsymbol{\mathcal{E}}$. In the case of a parametric right-hand side, this is straight-forward and the space $\mathcal{U}$ is a standard Bochner space, see Proposition 9. For varying domains or a varying exponent as a parametric dependency, the corresponding function spaces are intricate, we refer to Appendix C.

Next, we need to guarantee that the minimizer of $\boldsymbol{\mathcal{E}}$ indeed solves the parametric problem. This is carried out in Proposition 9 and is encoded in the fact that $\boldsymbol{u}^*(\boldsymbol{\tau}, \cdot) \in W^{1,p}(\Omega)$ for a.e. $\boldsymbol{\tau} \in \mathcal{P}$ minimizes $E_{\boldsymbol{\tau}}$ in the notation of this Proposition.

Proceeding to derive error estimates, we want to mimic the strategy of the non-parametric case. This crucially relies on the fact that the constants in equation 7 do not depend on the right-hand side. As a consequence, we can prove a two-sided estimate as in Proposition 9 with an analogue of $\rho_F^2 : W^{1,p}(\Omega) \times W^{1,p}(\Omega) \to \mathbb{R}$ given by $\rho_{\boldsymbol{\mathcal{F}}}^2 : L^p(\mathcal{P}, W^{1,p}(\Omega)) \times L^p(\mathcal{P}, W^{1,p}(\Omega)) \to \mathbb{R}$, for every $\boldsymbol{v}, \boldsymbol{w} \in L^p(\mathcal{P}, W^{1,p}(\Omega))$ defined by

$$\rho_{\boldsymbol{\mathcal{F}}}^2(\boldsymbol{v}, \boldsymbol{w}) := \int_{\mathcal{P}} \rho_F^2(\boldsymbol{v}(\boldsymbol{\tau}, \cdot), \boldsymbol{u}(\boldsymbol{\tau}, \cdot))\, \mathrm{d}\boldsymbol{\tau}\,.$$

Finally, we can proceed as in the non-parametric case and derive a Céa Lemma.

**Proposition 9** (Variable Right-Hand Sides). *Let $\Omega \subseteq \mathbb{R}^{d_\Omega}$, $d_\Omega \in \mathbb{N}$, and $\mathcal{P} \subseteq \mathbb{R}^{d_\mathcal{P}}$, $d_\mathcal{P} \in \mathbb{N}$, be bounded domains and $p \in (1, \infty)$. Assume $U \subset W^{1,p}(\Omega)$ is a closed subset that satisfies a Poincaré inequality, as in Proposition 2. Moreover, we define the Bochner–Lebesgue space*

$$\boldsymbol{\mathcal{U}} := L^p(\mathcal{P}, U)\,.$$

*For fixed $\boldsymbol{f} \in L^{p'}(\mathcal{P} \times \Omega)$, we define the variable right-hand side $p$-Dirichlet energy $\boldsymbol{\mathcal{E}} : \boldsymbol{\mathcal{U}} \to \mathbb{R}$ for every $\boldsymbol{v} \in \boldsymbol{\mathcal{U}}$ by*

$$\boldsymbol{\mathcal{E}}(\boldsymbol{v}) := \int_{\mathcal{P}} \left[ \frac{1}{p} \int_{\Omega} |\nabla_x \boldsymbol{v}(\boldsymbol{\tau}, \cdot)|^p\, \mathrm{d}x - \int_{\Omega} \boldsymbol{f}(\boldsymbol{\tau}, \cdot)\, \boldsymbol{v}(\boldsymbol{\tau}, \cdot)\, \mathrm{d}x \right] \mathrm{d}\boldsymbol{\tau}\,,$$

*where the gradient $\nabla_x$ for a.e. $\boldsymbol{\tau} \in \mathcal{P}$ is to be understood with respect to the variable $x \in \Omega$ only. Then, the following statements apply:*

*(i) There exists a unique (parametric) minimizer $\boldsymbol{u}^* \in \boldsymbol{\mathcal{U}}$ of $\boldsymbol{\mathcal{E}} : \boldsymbol{\mathcal{U}} \to \mathbb{R}$.*

*(ii) For a.e. $\boldsymbol{\tau} \in \mathcal{P}$, the function $\boldsymbol{u}^*(\boldsymbol{\tau}, \cdot) \in U$ is the unique minimizer of $E_{\boldsymbol{\tau}} : U \to \mathbb{R}$, for every $v \in U$ defined by*

$$E_{\boldsymbol{\tau}}(v) := \frac{1}{p} \int_{\Omega} |\nabla v|^p\, \mathrm{d}x - \int_{\Omega} \boldsymbol{f}(\boldsymbol{\tau}, \cdot)\, v\, \mathrm{d}x\,.$$

*(iii) Furthermore, for every $\boldsymbol{v} \in \boldsymbol{\mathcal{U}}$, it holds*

$$c(p)^{-1} \rho_{\boldsymbol{\mathcal{F}}}^2(\boldsymbol{v}, \boldsymbol{u}^*) \leq \boldsymbol{\mathcal{E}}(\boldsymbol{v}) - \boldsymbol{\mathcal{E}}(\boldsymbol{u}^*) \leq c(p)\, \rho_{\boldsymbol{\mathcal{F}}}^2(\boldsymbol{v}, \boldsymbol{u}^*)\,, \tag{9}$$

*where $c(p) > 0$ is the constant from Proposition 7 (ii).*

*Proof.* We prove this for more general parametric dependencies in the Appendix, see C. $\qquad\square$

**Remark 10.** Requiring $\boldsymbol{f}(\boldsymbol{\tau}, \cdot)$ to be mean-value-free for a.e. $\boldsymbol{\tau} \in \mathcal{P}$, we may, again, drop the assumption of a Poincaré inequality on the space $U$, as we explained in Remark 4. In this case, we cannot expect the minimizer to be unique.

The estimate 9 is the key to establish a Céa type Lemma for the energy $\boldsymbol{\mathcal{E}} : \boldsymbol{\mathcal{U}} \to \mathbb{R}$. In the situation of Proposition 9, we can accomplish this as in Lemma 5. More precisely, for any fixed $\boldsymbol{v} \in M \subset \boldsymbol{\mathcal{U}}$, it holds

$$\rho_{\boldsymbol{\mathcal{F}}}^2(\boldsymbol{v}, \boldsymbol{u}^*) \leq c(p) \left( \delta(\boldsymbol{v}) + \inf_{\tilde{\boldsymbol{v}} \in M} \rho_{\boldsymbol{\mathcal{F}}}^2(\tilde{\boldsymbol{v}}, \boldsymbol{u}^*) \right)\,, \tag{10}$$

where $\delta(\boldsymbol{v}) := \boldsymbol{\mathcal{E}}(\boldsymbol{v}) - \inf_{\tilde{\boldsymbol{v}} \in M} \boldsymbol{\mathcal{E}}(\tilde{\boldsymbol{v}})$ the parametric optimization error. With all the previous work, the Main Theorem is can now be proved in a similar way as in the case without parameters. We postpone the proof to the Appendix, see A.2.

## 3 NUMERICAL EXAMPLES

We give two examples, one with a high dimensional physical domain and one with a high-dimensional parametric right-hand side. Our goal is to investigate the error in dependence on the dimension. To find good neural network approximations, we employ a Deep Ritz Method for training.

**Example 1** As a PDE posed on a high-dimensional physical domain, we consider

$$-\Delta u + u = f \quad \text{in } \Omega,$$

with homogeneous Neumann boundary conditions, and $\Omega = (0,1)^{d_\Omega}$, $d_\Omega \in \mathbb{N}$. As manufactured solution, we use $u \in W^{1,2}(\Omega)$, for every $x = (x_1, \ldots, x_{d_\Omega})^\top \in \Omega$ defined by

$$u(x) := c \cdot \sum_{i=1}^{d_\Omega} \cos(\pi x_i),$$

where $c > 0$ is chosen such that $u$ has uniform $L^2(\Omega)$ norm. Then, the right-hand side $f \in L^2(\Omega)$, for every $x \in \Omega$ is given via

$$f(x) = (\pi^2 + 1)\, u(x).$$

**Example 2** As a problem with a high-dimensional parametric right-hand side, we consider

$$-\boldsymbol{u}'' + \boldsymbol{u} = \boldsymbol{f} \quad \text{in } \mathcal{P} \times \Omega,$$

with homogeneous Neumann boundary conditions, parameter space $\mathcal{P} = [-1,1]^{d_\mathcal{P}}$, $d_\mathcal{P} \in \mathbb{N}$, and physical domain $\Omega = (0,1)$. As manufactured solution and right-hand side, we use $\boldsymbol{u} \in L^2(\mathcal{P}, W^{1,2}(\Omega))$ and $\boldsymbol{f} \in L^2(\mathcal{P} \times \Omega)$, for every $(\boldsymbol{\tau}, x)^\top = (\tau_1, \ldots, \tau_{d_\mathcal{P}}, x)^\top \in \mathcal{P} \times \Omega$ defined by

$$\boldsymbol{u}(\boldsymbol{\tau}, x) := \sum_{k=0}^{d_\mathcal{P}-1} \frac{\tau_k}{k^2\pi^2 + 1} \cos(k\pi x) \quad \text{and} \quad \boldsymbol{f}(\boldsymbol{\tau}, x) = \sum_{k=0}^{d_\mathcal{P}-1} \tau_k \cos(k\pi x).$$

**Neural Network Architecture and Training** We employ fully-connected $\mathrm{ReLU}^2$-networks with four hidden layers and varying width as well as a Deep Ritz energy formulation as a loss function. To resolve the minimization problem, we employ the Adam optimizer with learning rate set to $0.001$. The appearing integrals are discretized using Monte–Carlo approximations, where new random points are drawn for every update in the gradient descent. The optimization is run until no further improvement is seen in approximating the ground truth, which in our examples happens typically in around $10,000$ to $20,000$ iterations.

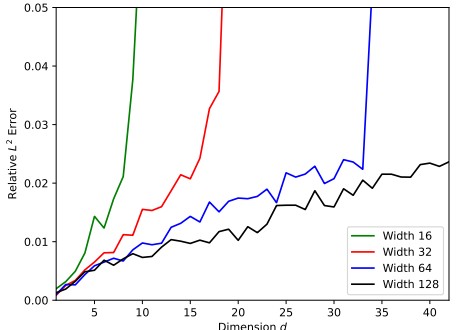 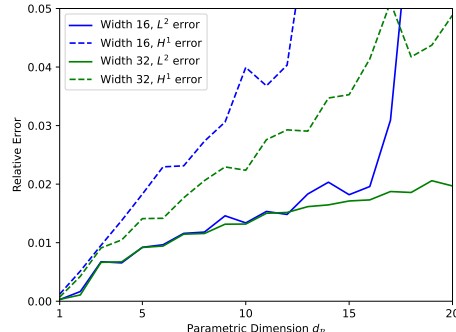

Figure 1: The plot on the left shows the relative $L^2$ errors obtained from Example 1 and the plot on the right reports the errors for Example 2. Here, dashed lines represent relative $H^1$ errors and solid lines stand for relative $L^2$ errors.

**Discussion** In Figure 1, we report the relative errors obtained for the two examples in dependence on the total dimension $d = d_\Omega + d_\mathcal{P} \in \mathbb{N}$ of the parametric cylinder $\mathcal{P} \times \Omega \subset \mathbb{R}^{d_\mathcal{P}} \times \mathbb{R}^{d_\Omega}$. In both examples, no exponential increase of the error is observable – at least for moderately high dimensions. Although it is impossible for us to quantify how well the empirically found solutions resolve the minimization, i.e., how large the quantity $\delta(u_\theta)$ in Theorem 1 is, the experiments still confirm the promising behavior of neural networks for solving high-dimensional and parametric problems.

AUTHOR CONTRIBUTIONS

All authors contributed equally to all parts of the manuscript.

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

# A  PROOFS

Here, we collect the proofs that were deferred to the Appendix.

## A.1  PROOF OF PROPOSITION 2

In order to prove Proposition 2, we need some preparation. The first two lemmas analyze the point-wise properties of the function $F : \mathbb{R}^d \to \mathbb{R}^d$, for every $a \in \mathbb{R}^d$ defined by $F(a) = |a|^{p-2}a$, that induces the natural distance measure $\rho_F^2 : W^{1,p}(\Omega) \times W^{1,p}(\Omega) \to \mathbb{R}$, which is essential for the error analysis of the $p$-Laplacian.

**Lemma 11.** *Let $p \in (1, \infty)$ and $d \in \mathbb{N}$. Then, there exists a constant $c(p) > 0$, depending only on $p \in (1, \infty)$ and not depending on $d \in \mathbb{N}$, such that the following statements apply:*

**(i)** *For every $a, b \in \mathbb{R}^d$, it holds*

$$c(p)^{-1} |F(a) - F(b)|^2 \le (|a|^{p-2}a - |b|^{p-2}b) \cdot (a - b) \le c(p) |F(a) - F(b)|^2.$$

**(ii)** *For every $a, b \in \mathbb{R}^d$, it holds*

$$c(p)^{-1} |F(a) - F(b)|^2 \le (|a| + |b|)^{p-2}|a - b|^2 \le c(p) |F(a) - F(b)|^2.$$

*Moreover, we can choose $c(p) > 0$ such that $(p \mapsto c(p)) \in C^0(1, \infty)$.*

*Proof.* See (Diening et al., 2007, Appendix) or (Diening & Ettwein, 2008, Appendix). Furthermore, carefully reviewing the proofs in (Diening et al., 2007, Appendix) reveals that the constants $c(p) > 0$, $p \in (1, \infty)$, in Lemma 11 depend continuously on $p \in (1, \infty)$. ☐

**Lemma 12.** *Let $p \in (1, \infty)$ and $d \in \mathbb{N}$. Then, there exits a constant $c(p) > 0$, depending only on $p \in (1, \infty)$ and not depending on $d \in \mathbb{N}$, such that for every $a, b \in \mathbb{R}^d$ with $|a| + |b| > 0$, we have that*

$$c(p)^{-1} |F(a) - F(b)|^2 \le \int_0^1 D^2\phi(\tau a + (1 - \tau)b) : (a - b) \otimes (a - b) (1 - \tau) \, d\tau$$
$$\le c(p) |F(a) - F(b)|^2,$$

*where $\phi \in C^1(\mathbb{R}^d) \cap C^2(\mathbb{R}^d \setminus \{0\})$, defined by $\phi(a) := \frac{1}{p}|a|^p$ for all $a \in \mathbb{R}^d$, denotes the $p$-Dirichlet density. Moreover, we can choose $c(p) > 0$ such that $(p \mapsto c(p)) \in C^0(1, \infty)$.*

*Proof.* We introduce the abbreviation $\eta^2 : \mathbb{R}^d \times \mathbb{R}^d \setminus \{(0,0)^\top\} \to \mathbb{R}_{\ge 0}$, for every $a, b \in \mathbb{R}^d$ with $|a| + |b| > 0$ defined by

$$\eta^2(a, b) := \int_0^1 D^2\phi(\tau a + (1 - \tau)b) : (a - b) \otimes (a - b) (1 - \tau) \, d\tau.$$

Using $D^2\phi(a) : b \otimes b \ge \min\{1, p - 1\}|a|^{p-2}|b|^2$ for all $a \in \mathbb{R}^d \setminus \{0\}$, $b \in \mathbb{R}^d$ (cf. (Růžička, 2004, p. 73, ineq. (1.35))), for every $a, b \in \mathbb{R}^d$ with $|a| + |b| > 0$, we obtain

$$\eta^2(a, b) \ge \min\{1, p - 1\} \int_0^1 |\tau a + (1 - \tau)b|^{p-2}|a - b|^2 (1 - \tau) \, d\tau. \tag{11}$$

With the help of Jensen's inequality applied with respect to the measure $d\mu = (1-\tau)d\tau$, i.e., in particular, we use that $d\mu([0,1]) = \frac{1}{2}$, for every $a, b \in \mathbb{R}^d$ with $|a| + |b| > 0$, we observe that

$$\left(2 \int_0^1 |\tau a + (1-\tau)b|(1-\tau)\, d\tau\right)^p \leq \int_0^1 |\tau a + (1-\tau)b|^p (1-\tau)\, d\tau. \tag{12}$$

Then, we continue in equation 11 by incorporating equation 12 and, thus, find that for every $a, b \in \mathbb{R}^d$ with $|a| + |b| > 0$, it holds

$$\begin{aligned}
\eta^2(a,b) &\geq \min\{1, p-1\} \int_0^1 |\tau a + (1-\tau)b|^p (1-\tau)\, d\tau \frac{|a-b|^2}{(|a|+|b|)^2} \\
&\geq \min\{1, p-1\} \left(2 \int_0^1 |\tau a + (1-\tau)b| (1-\tau)\, d\tau\right)^p \frac{|a-b|^2}{(|a|+|b|)^2}.
\end{aligned} \tag{13}$$

For every $a, b \in \mathbb{R}^d$, it holds

$$2 \int_0^1 |\tau a + (1-\tau)b| (1-\tau)\, d\tau \geq \frac{1}{6}(|a|+|b|), \tag{14}$$

which is based on that for $|a| > |b|$ and $\tau \in [\frac{2}{3}, 1]$, it holds $|\tau a + (1-\tau)b| \geq \frac{1}{3}|a| > \frac{1}{6}(|a|+|b|)$, and for $|b| \geq |a|$ and $\tau \in [0, \frac{1}{3}]$, it holds $|\tau a + (1-\tau)b| \geq \frac{1}{3}|b| \geq \frac{1}{6}(|a|+|b|)$. Using equation 14 in equation 13, for every $a, b \in \mathbb{R}^d$ with $|a| + |b| > 0$, we deduce that

$$\eta^2(a,b) \geq \min\{1, p-1\} \frac{1}{6^p}(|a|+|b|)^{p-2}|a-b|^2.$$

Resorting to Lemma 11, we conclude the existence of a constant $c(p) > 0$, depending only on $p \in (1, \infty)$, with $(p \mapsto c(p)) \in C^0(1, \infty)$, such that for every $a, b \in \mathbb{R}^d$ with $|a| + |b| > 0$, it holds

$$\eta^2(a,b) \geq c(p)^{-1}|F(a) - F(b)|^2.$$

On the other hand, since also $D^2\phi(a) : b \otimes b \leq \max\{1, p-2\}|a|^{p-2}|b|^2$ for all $a \in \mathbb{R}^d \setminus \{0\}$, $b \in \mathbb{R}^d$, which, again, follows very similarly to (Růžička, 2004, p. 73, ineq. (1.35)), we find that

$$\eta^2(a,b) \leq \max\{1, p-2\} \int_0^1 |\tau a + (1-\tau)b|^{p-2} (1-\tau)\, d\tau |a-b|^2. \tag{15}$$

Since, appealing to (Diening et al., 2007, Lemma 6.1), there is a constant $c(p) > 0$, depending only on $p \in (1, \infty)$, with $(p \mapsto c(p)) \in C^0(1, \infty)$, such that for every $a, b \in \mathbb{R}^d$ with $|a| + |b| > 0$, it holds

$$\int_0^1 |\tau a + (1-\tau)v|^{p-2}\, d\tau \leq c(p)(|a|+|b|)^{p-2},$$

we deduce from equation 15 that $\eta^2(a,b) \leq \max\{1, p-2\}c(p)(|a|+|b|)^{p-2}|a-b|^2$ for all $a, b \in \mathbb{R}^d$ with $|a| + |b| > 0$, which, resorting again to Lemma 11, completes the proof of Lemma 12. $\qquad\square$

*Proof of Proposition 2.* **ad (i).** The $p$-Dirichlet energy $E : U \to \mathbb{R}$ is proper, strictly convex, and continuous, hence, weakly lower semi-continuous. In addition, the validity of Poincaré's inequality guarantees the coercivity of $E : U \to \mathbb{R}$, so that the direct method in the calculus of variations yields, cf. Dacorogna (2007), the existence of a unique minimizer $u^* \in U$ of $E : U \to \mathbb{R}$.

**ad (ii).** We proceed similar to (Diening & Kreuzer, 2008, Lemma 16.). Again, we employ the notation $\phi \in C^1(\mathbb{R}^d) \cap C^2(\mathbb{R}^d \setminus \{0\})$, defined by $\phi(a) := \frac{1}{p}|a|^p$ for all $a \in \mathbb{R}^d$, for the $p$-Dirichlet density. Since $D\phi \in C^0(\mathbb{R}^d)^d$ with $|D\phi(a)| = |a|^{p-1}$ for all $a \in \mathbb{R}^d$, the $p$-Dirichlet energy is continuously Fréchet differentiable with

$$\langle DE(u), v \rangle_U := \int_\Omega D\phi(\nabla u) \cdot \nabla v\, dx - \langle f, v \rangle_{W^{1,p}(\Omega)}.$$

for all $u, v \in U$. In particular, due to the minimality of $u^* \in U$, we have that $DE(u^*) = 0$ in $U^*$, i.e., for every $v \in U$, it holds

$$\langle DE(u), v \rangle_U = 0. \tag{16}$$

However, $E : U \to \mathbb{R}$ is not twice continuously Fréchet differentiable. Therefore, we consider regularizations $(\phi_\varepsilon)_{\varepsilon > 0} \subseteq C^2(\mathbb{R}^d)$, defined by $\phi_\varepsilon(a) := \frac{1}{p}(\varepsilon^2 + |a|^2)^{\frac{p}{2}}$ for every $\varepsilon > 0$ and $a \in \mathbb{R}^d$, having the following properties:

($\alpha$) $\phi_\varepsilon(a) \to \phi(a)$ $(\varepsilon \to 0)$ for all $a \in \mathbb{R}^d$ and $\phi_\varepsilon(a) \le 2^{\frac{p}{2}}/p\,(|a|^p + \varepsilon^p)$ for all $a \in \mathbb{R}^d$ and $\varepsilon > 0$,

($\beta$) $(D\phi_\varepsilon)(a) \to (D\phi)(a)$ $(\varepsilon \to 0)$ for all $a \in \mathbb{R}^d$ and $|(D\phi_\varepsilon)(a)| \le 2^{\frac{p-1}{2}}(|a|^{p-1} + \varepsilon^{p-1})$ for all $a \in \mathbb{R}^d$ and $\varepsilon > 0$,

($\gamma$) $(D^2\phi_\varepsilon)(a) \to (D^2\phi)(a)$ $(\varepsilon \to 0)$ for all $a \in \mathbb{R}^d \setminus \{0\}$ and $|(D^2\phi_\varepsilon)(a)| \le (p-1)\,2^{\frac{p-2}{2}}(\varepsilon^{p-2} + |a|^{p-2})$ for all $a \in \mathbb{R}^d$ and $\varepsilon > 0$.

Inasmuch as $(\phi_\varepsilon)_{\varepsilon > 0} \subseteq C^2(\mathbb{R}^d)$ satisfies ($\alpha$), ($\beta$) and ($\gamma$), it is easily checked that for every $\varepsilon > 0$, the regularized $p$-Dirichlet energy $E^\varepsilon : U \to \mathbb{R}$, for every $v \in U$ defined by

$$E^\varepsilon(v) := \int_\Omega \phi_\varepsilon(\nabla v)\,\mathrm{d}x - \langle f, v\rangle_{W^{1,p}(\Omega)}\,,$$

is twice continuously Fréchet differentiable. In consequence, using Taylor's formula and Fubini's theorem, for every $\varepsilon > 0$ and $v \in U$, we obtain

$$E^\varepsilon(v) - E^\varepsilon(u^*) = \langle DE^\varepsilon(u^*), v - u^*\rangle_U$$
$$+ \int_0^1 D^2 E^\varepsilon(\tau v + (1-\tau)u^*)\,[v - u^*, v - u^*]\,(1-\tau)\,\mathrm{d}\tau \tag{17}$$
$$= \int_\Omega D\phi_\varepsilon(\nabla u^*) \cdot \nabla(v - u^*)\,\mathrm{d}x$$
$$+ \int_0^1 \int_\Omega D^2\phi_\varepsilon(\tau\nabla v + (1-\tau)\nabla u^*) : \nabla(v - u^*) \otimes \nabla(v - u^*)\,\mathrm{d}x\,(1-\tau)\,\mathrm{d}\tau$$
$$= \int_\Omega D\phi_\varepsilon(\nabla u^*) \cdot \nabla(v - u^*)\,\mathrm{d}x$$
$$+ \int_\Omega \int_0^1 D^2\phi_\varepsilon(\tau\nabla v + (1-\tau)\nabla u^*) : \nabla(v - u^*) \otimes \nabla(v - u^*)\,\mathrm{d}x\,(1-\tau)\,\mathrm{d}\tau\,.$$

Next, given both ($\alpha$), ($\beta$) and ($\gamma$), it is allowed to apply Lebesgue's dominated convergence theorem in equation 17. Hence, by passing for $\varepsilon \to 0$ in equation 17, using equation 16 in doing so, for every $v \in U$, we find that

$$E(v) - E(u^*) = \int_\Omega D\phi(\nabla u^*) \cdot \nabla(v - u^*)\,\mathrm{d}x \tag{18}$$
$$+ \int_\Omega \int_0^1 D^2\phi(\tau\nabla v + (1-\tau)\nabla u^*) : \nabla(v - u^*) \otimes \nabla(v - u^*)\,(1-\tau)\,\mathrm{d}\tau\,\mathrm{d}x$$
$$= \langle DE(u^*), v - u^*\rangle_U$$
$$+ \int_\Omega \int_0^1 D^2\phi(\tau\nabla v + (1-\tau)\nabla u^*) : \nabla(v - u^*) \otimes \nabla(v - u^*)\,(1-\tau)\,\mathrm{d}\tau\,\mathrm{d}x$$
$$= \int_\Omega \int_0^1 D^2\phi(\tau\nabla v + (1-\tau)\nabla u^*) : \nabla(v - u^*) \otimes \nabla(v - u^*)\,(1-\tau)\,\mathrm{d}\tau\,\mathrm{d}x\,.$$

Apart from that, resorting to Lemma 12, we deduce the existence of a constant $c(p) > 0$, depending only on $p \in (1, \infty)$, with $(p \mapsto c(p)) \in C^0(1, \infty)$, such that for every $v \in U$, it holds

$$c(p)^{-1}\rho_F^2(v, u^*) \le \int_\Omega \int_0^1 D^2\phi(\tau\nabla v + (1-\tau)\nabla u^*) : \nabla(v - u^*) \otimes \nabla(v - u^*)\,(1-\tau)\,\mathrm{d}\tau\,\mathrm{d}x$$
$$\le c(p)\,\rho_F^2(v, u^*)\,. \tag{19}$$

Eventually, by combining equation 18 and equation 19, we conclude the assertion of Theorem 2. $\square$

### A.2 THE REMAINING PROOFS

*Proof of Lemma 7.* The following proof is inspired by (Nakov & Touloupoulos, 2021, Section 3.1).

**ad (i)** By referring to Lemma 11 (ii), we deduce the existence of a constant $c(p) > 0$, depending only on $p \in (1, \infty)$, with $(p \mapsto c(p)) \in C^0(1, \infty)$, such that for every $u, v \in W^{1,p}(\Omega)$, it holds

$$\|\nabla u - \nabla v\|_{L^p(\Omega)^d}^p \le \int_\Omega |\nabla u - \nabla v|^2(|\nabla u| + |\nabla v|)^{p-2}\,\mathrm{d}x \le c(p)\,\rho_F^2(u, v)\,,$$

and, using Hölder's inequality with respect to $\left(\frac{p}{2}, \frac{p}{p-2}\right)$,

$$
\begin{aligned}
c(p)^{-1} \rho_F^2(u, v) &\leq \int_\Omega |\nabla u - \nabla v|^2 (|\nabla u| + |\nabla v|)^{p-2} \, \mathrm{d}x \\
&\leq \left( \int_\Omega |\nabla u - \nabla v|^p \, \mathrm{d}x \right)^{\frac{2}{p}} \left( \int_\Omega (|\nabla u| + |\nabla v|)^p \, \mathrm{d}x \right)^{\frac{p-2}{p}} \\
&\leq \left( \|\nabla u\|_{L^p(\Omega)^d} + \|\nabla v\|_{L^p(\Omega)^d} \right)^{p-2} \|\nabla u - \nabla v\|_{L^p(\Omega)^d}^2 .
\end{aligned}
$$

**ad (ii)** By referring to Lemma 11 (ii), we deduce the existence of a constant $c(p) > 0$, depending only on $p \in (1, \infty)$, with $(p \mapsto c(p)) \in C^0(1, \infty)$, such that for every $u, v \in W^{1,p}(\Omega)$, using Hölder's inequality with respect to $\left(\frac{2}{p}, \frac{2}{2-p}\right)$, it holds

$$
\begin{aligned}
\|\nabla(u - v)\|_{L^p(\Omega)^d} &\leq \left( \int_\Omega |\nabla(u - v)|^2 (|\nabla u| + |\nabla v|)^{p-2} \, \mathrm{d}x \right)^{\frac{p}{2}} \left( \int_\Omega (|\nabla u| + |\nabla v|)^p \, \mathrm{d}x \right)^{\frac{2-p}{2}} \\
&\leq \left( \|\nabla u\|_{L^p(\Omega)^d} + \|\nabla v\|_{L^p(\Omega)^d} \right)^{\frac{2p-p^2}{2}} \left( \int_\Omega |\nabla(u - v)|^2 (|\nabla u| + |\nabla v|)^{p-2} \, \mathrm{d}x \right)^{\frac{p}{2}} \\
&\leq c(p) \left( \|\nabla u\|_{L^p(\Omega)^d} + \|\nabla v\|_{L^p(\Omega)^d} \right)^{\frac{p(2-p)}{2}} \rho_F^2(u, v)^{\frac{p}{2}} ,
\end{aligned}
$$

and

$$
\begin{aligned}
c(p)^{-1} \rho_F^2(u, v) &\leq \int_\Omega |\nabla u - \nabla v|^2 (|\nabla u| + |\nabla v|)^{p-2} \, \mathrm{d}x \\
&\leq \int_\Omega |\nabla u - \nabla v|^p \frac{|\nabla u - \nabla v|^{2-p}}{(|\nabla u| + |\nabla v|)^{2-p}} \, \mathrm{d}x \leq \|\nabla u - \nabla v\|_{L^p(\Omega)^d}^p . \qquad \square
\end{aligned}
$$

**Lemma 13** (Coercivity of the $p$-Dirichlet Energy). *Let $f \in W^{1,p}(\Omega)^*$, $p \in (1, \infty)$, be such that $\langle f, c \rangle_{W^{1,p}(\Omega)} = 0$ for all $c \in \mathbb{R}$. Moreover, we define $E : W^{1,p}(\Omega) \to \mathbb{R}$ for every $v \in W^{1,p}(\Omega)$ by*

$$
E(v) := \frac{1}{p} \int_\Omega |\nabla v|^p \, \mathrm{d}x - \langle f, v \rangle_{W^{1,p}(\Omega)} . \tag{20}
$$

*Then, for every $v \in W^{1,p}(\Omega)$, we can estimate*

$$
\|\nabla v\|_{L^p(\Omega)^d} \leq c(p, \Omega) \left( \|f\|_{W^{1,p}(\Omega)^*}^{\frac{1}{p-1}} + E(v)^{\frac{1}{p}} \right) .
$$

*For a minimizer $u^* \in W^{1,p}(\Omega)$ of $E : W^{1,p}(\Omega) \to \mathbb{R}$, this reduces to*

$$
\|\nabla u^*\|_{L^p(\Omega)^d} \leq c(p, \Omega) \|f\|_{W^{1,p}(\Omega)^*}^{\frac{1}{p-1}} .
$$

*The constant $c(p, \Omega) > 0$ depends continuously on $p$ and on the domain $\Omega$.*

*Proof.* Using that $f \in W^{1,p}(\Omega)^*$ vanishes on constant functions, the Poincaré–Wirtinger inequality and the $\varepsilon$-Young inequality, for every $v \in W^{1,p}(\Omega)$ and $\varepsilon > 0$, abbreviating $\langle v \rangle_\Omega := \fint_\Omega v \, \mathrm{d}x$, it holds

$$
\begin{aligned}
E(v) &= \frac{1}{p} \|\nabla v\|_{L^p(\Omega)^d}^p + \langle f, v - \langle v \rangle_\Omega \rangle_{W^{1,p}(\Omega)} \\
&\geq \frac{1}{p} \|\nabla v\|_{L^p(\Omega)^d}^p - c(p, \varepsilon) \|f\|_{W^{1,p}(\Omega)^*}^{p'} - \varepsilon \|v - \langle v \rangle_\Omega\|_{W^{1,p}(\Omega)} \\
&\geq \left( \frac{1}{p} - \varepsilon \, C_P \right) \|\nabla v\|_{L^p(\Omega)^d}^p - c(p, \varepsilon) \|f\|_{W^{1,p}(\Omega)^*}^{p'} ,
\end{aligned} \tag{21}
$$

where $c(p, \varepsilon) := (p\varepsilon)^{1-p'} p^{-1}$. Hence, choosing $\varepsilon > 0$ sufficiently small – depending on the value of the Poincaré constant $C_P$ – in equation 21, for every $v \in W^{1,p}(\Omega)$, we find that

$$
\|\nabla v\|_{L^p(\Omega)^d} \leq c(p, \Omega) \left( E(v) + \|f\|_{W^{1,p}(\Omega)^*}^{p'} \right) , \tag{22}
$$

where the dependence of the Poincaré constant leads to dependence on the domain $\Omega$. $\qquad \square$

**Remark 14** (On the Constant $c(p, \Omega)$)**.** While the dependence of $c(p, \Omega)$ on $p$ can directly be understood from the proof, the dependence on $\Omega$ stems from the constant appearing in the Poincaré inequality – which we call the Poincaré constant and denote by $C_P$. For convex domains, one has that

$$C_P \leq \left(\frac{\pi_p}{\text{diam}(\Omega)}\right)^p, \quad \text{where} \quad \pi_p := 2\pi \frac{(p-1)^{\frac{1}{p}}}{p \sin(\pi/p)},$$

we refer to Esposito et al. (2013) or Koerber (2018).

Finally, we provide the missing proof of the Main Theorem.

*Proof of Theorem 1.* Let $p \geq 2$ and $\boldsymbol{v} \in \tilde{M}$ be arbitrary. Using Proposition 9 and the inequality 10, we estimate

$$\|\nabla_x \boldsymbol{v} - \nabla_x \boldsymbol{u}^*\|_{L^p(\mathcal{P} \times \Omega)^d}^p = \int_{\mathcal{P}} \|\nabla(\boldsymbol{v}(\boldsymbol{\tau}) - \boldsymbol{u}^*(\boldsymbol{\tau}))\|_{L^p(\Omega)^d}^p \, \mathrm{d}\boldsymbol{\tau}$$

$$\leq c(p) \int_{\mathcal{P}} \rho_F^2(\boldsymbol{v}(\boldsymbol{\tau}), \boldsymbol{u}^*(\boldsymbol{\tau})) \, \mathrm{d}\boldsymbol{\tau}$$

$$\leq c(p) \left(\delta(\boldsymbol{v}) + \inf_{\tilde{\boldsymbol{v}} \in \tilde{M}} \left[\int_{\mathcal{P}} \rho_F^2(\tilde{\boldsymbol{v}}(\boldsymbol{\tau}), \boldsymbol{u}^*(\boldsymbol{\tau})) \, \mathrm{d}\boldsymbol{\tau}\right]\right) =: (*).$$

We proceed by utilizing the relation of the natural distance $\rho_F^2 : W^{1,p}(\Omega) \times W^{1,p}(\Omega) \to \mathbb{R}$ to the Sobolev topology from Lemma 7 and, subsequently, apply Hölder's inequality with the exponents $(\frac{p}{p-2}, \frac{p}{2})$

$$(*) \leq \delta + \inf_{\tilde{\boldsymbol{v}} \in \tilde{M}} \left[\int_{\mathcal{P}} \left(\|\nabla \tilde{\boldsymbol{v}}(\boldsymbol{\tau})\|_{L^p(\Omega)^d} + \|\nabla \boldsymbol{u}^*(\boldsymbol{\tau})\|_{L^p(\Omega)^d}\right)^{p-2} \|\nabla \tilde{\boldsymbol{v}}(\boldsymbol{\tau}) - \nabla \boldsymbol{u}^*(\boldsymbol{\tau})\|_{L^p(\Omega)^d}^2 \mathrm{d}\boldsymbol{\tau}\right]$$

$$\leq \delta + \inf_{\tilde{\boldsymbol{v}} \in \tilde{M}} \left[\left(\int_{\mathcal{P}} \left(\|\nabla \tilde{\boldsymbol{v}}(\boldsymbol{\tau})\|_{L^p(\Omega)^d} + \|\nabla \boldsymbol{u}^*(\boldsymbol{\tau})\|_{L^p(\Omega)^d}\right)^p \mathrm{d}\boldsymbol{\tau}\right)^{\frac{p-2}{p}} \|\nabla_x \tilde{\boldsymbol{v}} - \nabla_x \boldsymbol{u}^*\|_{L^p(\mathcal{P} \times \Omega)^d}^2\right]$$

$$\leq \delta + 3^{p-2} \|\nabla_x \boldsymbol{u}^*\|_{L^p(\mathcal{P} \times \Omega)^d}^{p-2} \inf_{\tilde{\boldsymbol{v}} \in \tilde{M}} \|\nabla_x \tilde{\boldsymbol{v}} - \nabla_x \boldsymbol{u}^*\|_{L^p(\mathcal{P} \times \Omega)^d}^2$$

$$\leq \delta + 3^{p-2} c(p, \Omega) \|\boldsymbol{f}\|_{L^{p'}(\mathcal{P} \times \Omega)}^{\frac{p-2}{p-1}} \inf_{\tilde{\boldsymbol{v}} \in \tilde{M}} \|\nabla_x \tilde{\boldsymbol{v}} - \nabla_x \boldsymbol{u}^*\|_{L^p(\mathcal{P} \times \Omega)^d}^2.$$

This implies the assertion in the case $p \geq 2$. The proof in the situation of $p < 2$ works similarly and is therefore omitted. □

## B    SMOOTHNESS ASSUMPTION

For linear elliptic equations with parametric right-hand side, higher order Sobolev regularity holds true, provided the right-hand side $\boldsymbol{f} \in L^2(\mathcal{P} \times \Omega)$ and the domain $\Omega \subset \mathbb{R}^{d_\Omega}$, $d_\Omega \in \mathbb{N}$, are smooth enough. In the following, we denote by $H_f^k(\Omega)$, the Sobolev space with vanishing mean value.

**Lemma 15.** *Let $k \in \mathbb{N}$ be fixed and let $\mathcal{P} \subset \mathbb{R}^{d_\mathcal{P}}$, $d_\mathcal{P} \in \mathbb{N}$, be open and $\Omega \subset \mathbb{R}^{d_\Omega}$, $d_\Omega \in \mathbb{N}$, be a domain with $\partial\Omega \in C^{k+1,1}$ boundary. Let, furthermore, $\boldsymbol{f} \in C^{k+2}(\mathcal{P}, H_f^k(\Omega))$ be given. Then, the weak solution $\boldsymbol{u}^* \in L^2(\mathcal{P}, H_f^1(\Omega))$ to*

$$\begin{aligned} -\Delta \boldsymbol{u} &= \boldsymbol{f} \quad \text{in } \mathcal{P} \times \Omega, \\ \partial_n \boldsymbol{u} &= 0 \quad \text{on } \mathcal{P} \times \partial\Omega, \end{aligned} \tag{23}$$

*is a member of the space $H^{k+2}(\mathcal{P} \times \Omega)$.*

*Proof.* We define $e : H_f^{k+2}(\Omega) \times \mathcal{P} \to H_f^k(\Omega)$ for every $u \in H_f^{k+2}(\Omega)$ and $\boldsymbol{\tau} \in \mathcal{P}$ by

$$e(u, \boldsymbol{\tau}) := -\Delta u - f(\tau) \quad H_f^k(\Omega).$$

Then, the zero level set

$$\left\{(u, \boldsymbol{\tau})^\top \in H_f^{k+2}(\Omega) \times \mathcal{P} \mid e(u, \tau) = 0\right\}$$

is parametrized through the solution map $(\boldsymbol{\tau} \mapsto \boldsymbol{u}^*(\boldsymbol{\tau})) : \mathcal{P} \to H_f^{k+2}(\Omega)$. In fact, standard elliptic regularity theory yields that $\boldsymbol{u}^*(\boldsymbol{\tau}) \in H^{k+2}(\Omega)$ for a.e. $\boldsymbol{\tau} \in \mathcal{P}$, see for instance Grisvard (2011). The implicit function theorem for Banach spaces guarantees that $\boldsymbol{u}^* \in C^{k+2}(\mathcal{P}, H^{k+2}(\Omega))$ provided the partial derivative of $e$ with respect to the first component, i.e., $\partial_1 e(u, \tau) : H_f^{k+2}(\Omega) \to H_f^k(\Omega)$, for every $(u, \tau)^\top \in H_f^{k+2}(\Omega) \times \mathcal{P}$ given via

$$\partial_1 e(u, \tau)[v] = -\Delta v \quad \text{in } H_f^k(\Omega) \quad \text{for all } v \in H_f^{k+2}(\Omega),$$

is a linear homeomorphism. However, by using the exact same elliptic regularity result as used above, we see that this is in fact true. Hence, the assertion of the lemma follows. $\qquad\square$

## C    MORE GENERAL PARAMETRIC DEPENDENCIES

In the main part of the manuscript, we only considered parametric dependencies that were induced through a parameter dependent right-hand side. This was done to keep the technicality minimal, yet does not constitute the full generality of our analysis. In this Section, we outline more general parametric dependencies, including varying integrability exponents, domains and material tensors. In every situation, we guarantee the well-posedness of the parametric problem, providing an analogue result to Proposition 9 and, consequently, allows to deduce error decay rates. As error decay rates – assuming smoothness – follow the same pattern as in Theorem 1, we do not explicitly state them.

### C.1    PARAMETRIC EXPONENTS AND PARAMETRIC RIGHT-HAND SIDES

We begin with a problem where both, the exponent $p \in (1, \infty)$ and the right-hand side $\boldsymbol{f}$ are allowed to vary in a parameter space $\mathcal{P}$. More precisely, we seek $\boldsymbol{u}^* : \mathcal{P} \times \Omega \to \mathbb{R}$ satisfying

$$-\operatorname{div}\left(|\nabla_x \boldsymbol{u}^*(\boldsymbol{\tau}, x)|^{p(\boldsymbol{\tau})-2} \nabla_x \boldsymbol{u}^*(\boldsymbol{\tau}, x)\right) = \boldsymbol{f}(\boldsymbol{\tau}, x) \quad \text{for a.e. } (\boldsymbol{\tau}, x)^\top \in \mathcal{P} \times \Omega,$$

subjected to suitable boundary conditions. The precise statement is the following.

**Proposition 16** (Variable Exponents)**.** *Let $\Omega \subseteq \mathbb{R}^{d_\Omega}$, $d_\Omega \in \mathbb{N}$, and $\mathcal{P} \subseteq \mathbb{R}^{d_\mathcal{P}}$, $d_\mathcal{P} \in \mathbb{N}$, be bounded domains and $p \in L^\infty(\mathcal{P})$ such that there exist $p^-, p^+ \in (1, \infty)$ with $p^- \leq p(\boldsymbol{\tau}) \leq p^+$ for a.e. $\boldsymbol{\tau} \in \mathcal{P}$. Moreover, we define the variable exponent Lebesgue space[4]*

$$L^{p(\cdot)}(\mathcal{P} \times \Omega) := \left\{ \boldsymbol{v} \in L^0(\mathcal{P} \times \Omega) \ \middle| \ \int_{\mathcal{P}} \int_{\Omega} |\boldsymbol{v}(\boldsymbol{\tau}, x)|^{p(\boldsymbol{\tau})} \, \mathrm{d}x \, \mathrm{d}\boldsymbol{\tau} < \infty \right\},$$

*and the variable exponent Bochner–Lebesgue space*

$$\boldsymbol{\mathcal{U}} := \left\{ \boldsymbol{v} \in L^{p(\cdot)}(\mathcal{P} \times \Omega) \mid \boldsymbol{v}(\boldsymbol{\tau}, \cdot) \in W_0^{1, p(\boldsymbol{\tau})}(\Omega) \text{ for a.e. } \boldsymbol{\tau} \in \mathcal{P}, |\nabla_x \boldsymbol{v}| \in L^{p(\cdot)}(\mathcal{P} \times \Omega) \right\},$$

*where the gradient $\nabla_x$ for a.e. $\boldsymbol{\tau} \in \mathcal{P}$ is to be understood with respect to the variable $x \in \Omega$ only. For fixed $\boldsymbol{f} \in L^{p'(\cdot)}(\mathcal{P} \times \Omega)$, i.e., $\boldsymbol{f} \in L^0(\mathcal{P} \times \Omega)$ and $\int_{\mathcal{P}} \int_{\Omega} |\boldsymbol{f}(\boldsymbol{\tau}, x)|^{p'(\boldsymbol{\tau})} \, \mathrm{d}x \, \mathrm{d}\boldsymbol{\tau} < \infty$, where $p' \in L^\infty(\mathcal{P})$ is defined by $p'(\boldsymbol{\tau}) := \frac{p(\boldsymbol{\tau})}{p(\boldsymbol{\tau})-1}$ for a.e. $\boldsymbol{\tau} \in \mathcal{P}$, we define variable exponent $p(\cdot)$-Dirichlet energy $\boldsymbol{\mathcal{E}} : \boldsymbol{\mathcal{U}} \to \mathbb{R}$ for every $\boldsymbol{v} \in \boldsymbol{\mathcal{U}}$ by*

$$\mathcal{E}(\boldsymbol{v}) := \int_{\mathcal{P}} \left[ \frac{1}{p(\boldsymbol{\tau})} \int_{\Omega} |\nabla_x \boldsymbol{v}(\boldsymbol{\tau}, \cdot)|^{p(\boldsymbol{\tau})} \, \mathrm{d}x - \int_{\Omega} \boldsymbol{f}(\boldsymbol{\tau}, \cdot) \, \boldsymbol{v}(\boldsymbol{\tau}, \cdot) \, \mathrm{d}x \right] \mathrm{d}\boldsymbol{\tau}.$$

*Then, the following statements apply:*

*(i) There exists a unique (parametric) minimizer $\boldsymbol{u}^* \in \boldsymbol{\mathcal{U}}$ of $\boldsymbol{\mathcal{E}} : \boldsymbol{\mathcal{U}} \to \mathbb{R}$.*

*(ii) For a.e. $\boldsymbol{\tau} \in \mathcal{P}$, $\boldsymbol{u}^*(\boldsymbol{\tau}, \cdot) \in W_0^{1, p(\boldsymbol{\tau})}(\Omega)$ is a unique minimizer of $E_{\boldsymbol{\tau}} : W_0^{1, p(\boldsymbol{\tau})}(\Omega) \to \mathbb{R}$, for every $v \in W_0^{1, p(\boldsymbol{\tau})}(\Omega)$ defined by*

$$E_{\boldsymbol{\tau}}(v) := \frac{1}{p(\boldsymbol{\tau})} \int_{\Omega} |\nabla v|^{p(\boldsymbol{\tau})} \mathrm{d}x - \int_{\Omega} \boldsymbol{f}(\boldsymbol{\tau}, \cdot) \, v \, \mathrm{d}x.$$

---

[4]Here, $L^0(\mathcal{P} \times \Omega)$ denotes the space of scalar (Lebesgue–)measurable functions on $\mathcal{P} \times \Omega$.

*(iii)* For a.e. $\boldsymbol{\tau} \in \mathcal{P}$ and $v \in W_0^{1,p(\boldsymbol{\tau})}(\Omega)$, it holds

$$c(p(\boldsymbol{\tau}))^{-1} \left\| F_{\boldsymbol{\tau}}(\nabla v) - F_{\boldsymbol{\tau}}(\nabla_x \boldsymbol{u}^*(\boldsymbol{\tau}, \cdot)) \right\|_{L^2(\Omega)^d}^2 \leq E_{\boldsymbol{\tau}}(v) - E_{\boldsymbol{\tau}}(\boldsymbol{u}^*(\boldsymbol{\tau}, \cdot))$$
$$\leq c(p(\boldsymbol{\tau})) \left\| F_{\boldsymbol{\tau}}(\nabla v) - F_{\boldsymbol{\tau}}(\nabla_x \boldsymbol{u}^*(\boldsymbol{\tau}, \cdot)) \right\|_{L^2(\Omega)^d}^2 ,$$

where $F_{\boldsymbol{\tau}} : \mathbb{R}^d \to \mathbb{R}^d$, $\boldsymbol{\tau} \in \mathcal{P}$, for every $\boldsymbol{\tau} \in \mathcal{P}$ is defined by $F_{\boldsymbol{\tau}}(a) \coloneqq |a|^{\frac{p(\boldsymbol{\tau})-2}{2}} a$ for all $a \in \mathbb{R}^d$ and $c(p(\boldsymbol{\tau})) > 0$ is the constant from Theorem 2.

*(iv)* Furthermore, for every $\boldsymbol{v} \in \mathcal{U}$, it holds

$$\operatorname*{ess\,inf}_{\boldsymbol{\tau} \in \mathcal{P}} c(p(\boldsymbol{\tau}))^{-1} \, \boldsymbol{\rho}_{\boldsymbol{\mathcal{F}}}^2(\boldsymbol{v}, \boldsymbol{u}^*) \leq \boldsymbol{\mathcal{E}}(\boldsymbol{v}) - \boldsymbol{\mathcal{E}}(\boldsymbol{u}^*) \leq \operatorname*{ess\,sup}_{\boldsymbol{\tau} \in \mathcal{P}} c(p(\boldsymbol{\tau})) \, \boldsymbol{\rho}_{\boldsymbol{\mathcal{F}}}^2(\boldsymbol{v}, \boldsymbol{u}^*) ,$$

where

$$\boldsymbol{\rho}_{\boldsymbol{\mathcal{F}}}^2(\boldsymbol{v}, \boldsymbol{u}^*) \coloneqq \int_{\mathcal{P}} \left\| F_{\boldsymbol{\tau}}(\nabla_x \boldsymbol{v}(\boldsymbol{\tau}, \cdot)) - F_{\boldsymbol{\tau}}(\nabla_x \boldsymbol{u}^*(\boldsymbol{\tau}, \cdot)) \right\|_{L^2(\Omega)^d}^2 \mathrm{d}\boldsymbol{\tau} .$$

*Proof.* **ad (i).** The space $\mathcal{U}$ equipped with the norm $\| \cdot \|_{\mathcal{U}} \coloneqq \| \cdot \|_{L^{p(\cdot)}(\mathcal{P} \times \Omega)} + \| |\nabla_x \cdot| \|_{L^{p(\cdot)}(\mathcal{P} \times \Omega)}$, where

$$\|\boldsymbol{v}\|_{L^{p(\cdot)}(\mathcal{P} \times \Omega)} \coloneqq \inf \left\{ \lambda > 0 \;\Big|\; \int_{\mathcal{P}} \int_{\Omega} \left| \frac{\boldsymbol{v}(\boldsymbol{\tau}, x)}{\lambda} \right|^{p(\boldsymbol{\tau})} \mathrm{d}x \, \mathrm{d}\boldsymbol{\tau} \leq 1 \right\}$$

denotes the Luxembourg norm, cf. Diening et al. (2011), is a reflexive Banach space, cf. (Kaltenbach, 2021, Proposition 3.7 & Proposition 3.9) or (Kaltenbach & Růžička, 2021, Proposition 3.6 & Proposition 3.7)[5]. Apparently, $\boldsymbol{\mathcal{E}} : \mathcal{U} \to \mathbb{R}$ is strictly convex and continuous. In addition, for every $\boldsymbol{v} \in \mathcal{U}$, due to Poincaré's inequality applied for a.e. fixed $\boldsymbol{\tau} \in \mathcal{P}$, which is allowed since $\boldsymbol{v}(\boldsymbol{\tau}, \cdot) \in W_0^{1,p(\boldsymbol{\tau})}(\Omega)$ for a.e. $\boldsymbol{\tau} \in \mathcal{P}$, we have that

$$\int_{\mathcal{P}} \int_{\Omega} |\boldsymbol{v}(\boldsymbol{\tau}, x)|^{p(\boldsymbol{\tau})} \mathrm{d}x \, \mathrm{d}\boldsymbol{\tau} \leq \int_{\mathcal{P}} \left( 2 \operatorname{diam}(\Omega) \right)^{p(\boldsymbol{\tau})} \int_{\Omega} |\nabla_x \boldsymbol{v}(\boldsymbol{\tau}, x)|^{p(\boldsymbol{\tau})} \mathrm{d}x \, \mathrm{d}\boldsymbol{\tau}$$
$$\leq \left( 1 + 2\operatorname{diam}(\Omega) \right)^{p^+} \int_{\mathcal{P}} \int_{\Omega} |\nabla_x \boldsymbol{v}(\boldsymbol{\tau}, x)|^{p(\boldsymbol{\tau})} \mathrm{d}x \, \mathrm{d}\boldsymbol{\tau} , \tag{24}$$

which for every $\boldsymbol{v} \in \mathcal{U}$ and $\varepsilon \in (0, \frac{1}{p^-}]$, using for a.e. $\boldsymbol{\tau} \in \mathcal{P}$, the $\varepsilon$-Young inequality with constant $c(p(\boldsymbol{\tau}), \varepsilon) \coloneqq \frac{(p(\boldsymbol{\tau})\varepsilon)^{1-p'(\boldsymbol{\tau})}}{p'(\boldsymbol{\tau})}$, implies that

$$\boldsymbol{\mathcal{E}}(\boldsymbol{v}) \geq \int_{\mathcal{P}} \frac{1}{p(\boldsymbol{\tau})} \int_{\Omega} |\nabla_x \boldsymbol{v}(\boldsymbol{\tau}, \cdot)|^{p(\boldsymbol{\tau})} \mathrm{d}x \, \mathrm{d}\boldsymbol{\tau}$$
$$- \int_{\mathcal{P}} \int_{\Omega} c(p(\boldsymbol{\tau}), \varepsilon) |\boldsymbol{f}(\boldsymbol{\tau}, \cdot)|^{p'(\boldsymbol{\tau})} - \varepsilon |\boldsymbol{v}(\boldsymbol{\tau}, \cdot)|^{p(\boldsymbol{\tau})} \mathrm{d}x \, \mathrm{d}\boldsymbol{\tau}$$
$$\geq \left( \frac{1}{p^+} - \varepsilon(1 + 2\operatorname{diam}(\Omega))^{p^+} \right) \int_{\mathcal{P}} \int_{\Omega} |\nabla_x \boldsymbol{v}(\boldsymbol{\tau}, \cdot)|^{p(\boldsymbol{\tau})} \mathrm{d}x \, \mathrm{d}\boldsymbol{\tau} \tag{25}$$
$$- \frac{(p^- \varepsilon)^{1-(p^-)'}}{(p^+)'} \int_{\mathcal{P}} \int_{\Omega} |\boldsymbol{f}(\boldsymbol{\tau}, \cdot)|^{p'(\boldsymbol{\tau})} \mathrm{d}x \, \mathrm{d}\boldsymbol{\tau} .$$

Hence, since $\int_{\mathcal{P}} \int_{\Omega} |\boldsymbol{v}(\boldsymbol{\tau}, \cdot)|^{p(\boldsymbol{\tau})} + |\nabla_x \boldsymbol{v}(\boldsymbol{\tau}, \cdot)|^{p(\boldsymbol{\tau})} \mathrm{d}x \, \mathrm{d}\boldsymbol{\tau} \to \infty$ if $\|\boldsymbol{v}\|_{\mathcal{U}} \to \infty$ (cf. (Diening et al., 2011, Lemma 3.2.4)) from equation 24 and equation 25 for $\varepsilon \in (0, \frac{1}{p^-}]$ sufficiently small, we conclude that from $\|\boldsymbol{v}\|_{\mathcal{U}} \to \infty$, it follows that $\boldsymbol{\mathcal{E}}(\boldsymbol{v}) \to \infty$, i.e., $\boldsymbol{\mathcal{E}} : \mathcal{U} \to \mathbb{R}$ is weakly coercive, so that the direct method in the calculus of variations, cf. Dacorogna (2007), yields the existence of a unique minimizer $\boldsymbol{u}^* \in \mathcal{U}$ of $\boldsymbol{\mathcal{E}} : \mathcal{U} \to \mathbb{R}$.

**ad (ii).** A standard calculation shows that $\boldsymbol{\mathcal{E}} : \mathcal{U} \to \mathbb{R}$ is continuously Fréchet differentiable with

$$\langle D\boldsymbol{\mathcal{E}}(\boldsymbol{u}), \boldsymbol{v} \rangle_{\mathcal{U}} = \int_{\mathcal{P}} \langle DE_{\boldsymbol{\tau}}(\boldsymbol{u}(\boldsymbol{\tau}, \cdot)), \boldsymbol{v}(\boldsymbol{\tau}, \cdot) \rangle_{W_0^{1,p(\boldsymbol{\tau})}(\Omega)} \mathrm{d}\boldsymbol{\tau}$$

---

[5]More precisely, these references prove only the case $N = 1$, since therein $\mathcal{P}$ represents a time interval in an unsteady fluid flow problem. However, the proofs can be generalized verbatimly to the case $N > 1$, so that we will refrain from proving these results again at this point.

for all $\boldsymbol{u}, \boldsymbol{v} \in \mathcal{U}$. Therefore, due to the minimality of $\boldsymbol{u}^* \in \mathcal{U}$, for every $\boldsymbol{v} \in \mathcal{U}$, we have that

$$0 = \langle D\boldsymbol{\mathcal{E}}(\boldsymbol{u}^*), \boldsymbol{v} \rangle_{\mathcal{U}} = \int_{\mathcal{P}} \langle DE_{\boldsymbol{\tau}}(\boldsymbol{u}^*(\boldsymbol{\tau}, \cdot)), \boldsymbol{v}(\boldsymbol{\tau}, \cdot) \rangle_{W_0^{1,p(\boldsymbol{\tau})}(\Omega)} \, \mathrm{d}\boldsymbol{\tau} \,. \tag{26}$$

Inasmuch as $W_0^{1,p^+}(\Omega) \hookrightarrow W_0^{1,p(\boldsymbol{\tau})}(\Omega)$ densely for a.e. $\boldsymbol{\tau} \in \mathcal{P}$ and $W_0^{1,p^+}(\Omega)$ is separable and, thus, contains a countable dense subset $(\psi_k)_{k \in \mathbb{N}} \subseteq W_0^{1,p^+}(\Omega)$, the subset $(\psi_k)_{k \in \mathbb{N}}$ lies even densely in $W_0^{1,p(\boldsymbol{\tau})}(\Omega)$ for a.e. $\boldsymbol{\tau} \in \mathcal{P}$. Next, choosing $\boldsymbol{v} = \varphi \psi_k \in \mathcal{U}$ in equation 26 for arbitrary $\varphi \in C_0^{\infty}(\mathcal{P})$ and $k \in \mathbb{N}$, we further deduce that

$$\int_{\mathcal{P}} \langle DE_{\boldsymbol{\tau}}(\boldsymbol{u}^*(\boldsymbol{\tau}, \cdot)), \psi_k \rangle_{W_0^{1,p(\boldsymbol{\tau})}(\Omega)} \varphi(\boldsymbol{\tau}) \, \mathrm{d}\boldsymbol{\tau} = 0 \,, \tag{27}$$

so that for each fixed $k \in \mathbb{N}$, the fundamental lemma of calculus of variations implies that for a.e. $\boldsymbol{\tau} \in \mathcal{P}$, it holds $\langle DE_{\boldsymbol{\tau}}(\boldsymbol{u}^*(\boldsymbol{\tau}, \cdot)), \psi_k \rangle_{W_0^{1,p(\boldsymbol{\tau})}(\Omega)} = 0$. This, since the countable union of sets of zero measure has still zero measure, we deduce from equation 27 that for a.e. $\boldsymbol{\tau} \in \mathcal{P}$, it holds for all $k \in \mathbb{N}$

$$\langle DE_{\boldsymbol{\tau}}(\boldsymbol{u}^*(\boldsymbol{\tau}, \cdot)), \psi_k \rangle_{W_0^{1,p(\boldsymbol{\tau})}(\Omega)} = 0 \,. \tag{28}$$

Since $(\psi_k)_{k \in \mathbb{N}}$ is dense in $W_0^{1,p(\boldsymbol{\tau})}(\Omega)$ for a.e. $\boldsymbol{\tau} \in \mathcal{P}$, from equation 28 we infer that for a.e. $\boldsymbol{\tau} \in \mathcal{P}$, it holds for all $v \in W_0^{1,p(\boldsymbol{\tau})}(\Omega)$

$$\langle DE_{\boldsymbol{\tau}}(\boldsymbol{u}^*(\boldsymbol{\tau}, \cdot)), v \rangle_{W_0^{1,p(\boldsymbol{\tau})}(\Omega)} = 0 \,.$$

Eventually, since for a.e. $\boldsymbol{\tau} \in \mathcal{P}$, the $p(\boldsymbol{\tau})$-Dirichlet energy $E_{\boldsymbol{\tau}} : W_0^{1,p(\boldsymbol{\tau})}(\Omega) \to \mathbb{R}$ is strictly convex, for a.e. $\boldsymbol{\tau} \in \mathcal{P}$, the slice $\boldsymbol{u}^*(\boldsymbol{\tau}, \cdot) \in W_0^{1,p(\boldsymbol{\tau})}(\Omega)$ is a unique minimizer of $E_{\boldsymbol{\tau}} : W_0^{1,p(\boldsymbol{\tau})}(\Omega) \to \mathbb{R}$.

**ad (iii) and (iv).** Follows from point (ii) and Theorem 2. $\qquad\square$

## C.2 PARAMETRIC DOMAINS

We consider parametric domains, where we focus on domains depending on only one parameter, as the required function spaces are only studied in this case. More precisely, we aim to solve

$$-\operatorname{div}\left(|\nabla_x \boldsymbol{u}^*(\boldsymbol{\tau}, x)|^{p-2} \nabla \boldsymbol{u}_x(\boldsymbol{\tau}, x)\right) = \boldsymbol{f}(\boldsymbol{\tau}, x) \quad \text{for a.e. } x \in \Omega(\boldsymbol{\tau}), \, \boldsymbol{\tau} \in \mathcal{P} \,.$$

The precise requirements are given in the following proposition.

**Proposition 17** (Variable Domains). *Let $\Omega \subseteq \mathbb{R}^{d_\Omega}$, $d_\Omega \in \mathbb{N}$, a bounded Lipschitz domain and $p \in (1, \infty)$. Moreover, let $\varphi_{\boldsymbol{\tau}} : \Omega \to \Omega(\boldsymbol{\tau})$, $\boldsymbol{\tau} \in \mathcal{P} := (0, T)$, $T > 0$, the induced flow of a smooth, compactly supported vector field $\mathbf{v} : \mathbb{R} \times \mathbb{R}^d \to \mathbb{R}^d$, cf. (Delfour & Zolésio, 2011, Chapter 4). For the non-cylindrical domain $Q := \bigcup_{\boldsymbol{\tau} \in \mathcal{P}} \{\boldsymbol{\tau}\} \times \Omega(\boldsymbol{\tau})$, we define the variable domain Bochner–Lebesgue space*

$$\mathcal{U} := L^p(\mathcal{P}, W_0^{1,p}(\Omega(\cdot))) := \{\boldsymbol{u} \in L^p(Q) \mid \boldsymbol{u}(\boldsymbol{\tau}, \cdot) \in W_0^{1,p}(\Omega(\boldsymbol{\tau})) \text{ for a.e. } \boldsymbol{\tau} \in \mathcal{P}, |\nabla_x \boldsymbol{u}| \in L^p(Q)\} \,,$$

*where the gradient $\nabla_x$ for a.e. $\boldsymbol{\tau} \in \mathcal{P}$ is to be understood with respect to the variable $x \in \Omega(\boldsymbol{\tau})$ only. For fixed $\boldsymbol{f} \in L^{p'}(Q)$, we define the variable domain p-Dirichlet energy $\boldsymbol{\mathcal{E}} : \mathcal{U} \to \mathbb{R}$ for every $\boldsymbol{v} \in \mathcal{U}$ by*

$$\boldsymbol{\mathcal{E}}(\boldsymbol{v}) := \int_{\mathcal{P}} \left[ \frac{1}{p} \int_{\Omega(\boldsymbol{\tau})} |\nabla_x \boldsymbol{v}(\boldsymbol{\tau}, \cdot)|^p \, \mathrm{d}x - \int_{\Omega(\boldsymbol{\tau})} \boldsymbol{f}(\boldsymbol{\tau}, \cdot) \, \boldsymbol{v}(\boldsymbol{\tau}, \cdot) \, \mathrm{d}x \right] \mathrm{d}\boldsymbol{\tau} \,.$$

*Then, the following statements apply:*

*(i) There exists a unique (parametric) minimizer $\boldsymbol{u}^* \in \mathcal{U}$ of $\boldsymbol{\mathcal{E}} : \mathcal{U} \to \mathbb{R}$.*

*(ii) For a.e. $\boldsymbol{\tau} \in \mathcal{P}$, $\boldsymbol{u}^*(\boldsymbol{\tau}, \cdot) \in W_0^{1,p}(\Omega(\boldsymbol{\tau}))$ is a unique minimizer of $E_{\boldsymbol{\tau}} : W_0^{1,p}(\Omega(\boldsymbol{\tau})) \to \mathbb{R}$, for every $v \in W_0^{1,p}(\Omega(\boldsymbol{\tau}))$ defined by*

$$E_{\boldsymbol{\tau}}(v) := \frac{1}{p} \int_{\Omega(\boldsymbol{\tau})} |\nabla v|^p \, \mathrm{d}x - \int_{\Omega(\boldsymbol{\tau})} \boldsymbol{f}(\boldsymbol{\tau}, \cdot) \, v \, \mathrm{d}x \,.$$

*(iii) For a.e. $\boldsymbol{\tau} \in \mathcal{P}$ and $v \in W_0^{1,p(\boldsymbol{\tau})}(\Omega)$, it holds*

$$c(p)^{-1} \|F(\nabla v) - F(\nabla_x \boldsymbol{u}^*(\boldsymbol{\tau}, \cdot))\|_{L^2(\Omega(\boldsymbol{\tau}))^d}^2 \leq E_{\boldsymbol{\tau}}(v) - E_{\boldsymbol{\tau}}(\boldsymbol{u}^*(\boldsymbol{\tau}, \cdot))$$

$$\leq c(p) \|F(\nabla v) - F(\nabla_x \boldsymbol{u}^*(\boldsymbol{\tau}, \cdot))\|_{L^2(\Omega(\boldsymbol{\tau}))^d}^2 \,,$$

*where $c(p) > 0$ is the constant from Theorem 2.*

*(iv) Furthermore, for every $v \in \mathcal{U}$, it holds*

$$\operatorname*{ess\,inf}_{\boldsymbol{\tau} \in \mathcal{P}} c(p)^{-1} \, \rho_{\boldsymbol{\mathcal{F}}}^2(v, u^*) \le \boldsymbol{\mathcal{E}}(v) - \boldsymbol{\mathcal{E}}(u^*) \le \operatorname*{ess\,sup}_{\boldsymbol{\tau} \in \mathcal{P}} c(p) \, \rho_{\boldsymbol{\mathcal{F}}}^2(v, u^*) \,,$$

*where*

$$\rho_{\boldsymbol{\mathcal{F}}}^2(v, u^*) \coloneqq \int_{\mathcal{P}} \|F(\nabla_x v(\boldsymbol{\tau}, \cdot)) - F(\nabla_x u^*(\boldsymbol{\tau}, \cdot))\|_{L^2(\Omega(\boldsymbol{\tau}))^d}^2 \, \mathrm{d}\boldsymbol{\tau}.$$

*Proof.* **ad (i).** The space $\mathcal{U}$ equipped with the norm $\|\cdot\|_{\mathcal{U}} \coloneqq \|\cdot\|_{L^p(Q)} + \|\,|\nabla \cdot|\,\|_{L^p(Q)}$, forms a reflexive Banach space, cf. (Nägele, 2015, Proposition 3.17 & Corollary 3.25) or Nägele et al. (2017); Nägele & Růžička (2018). Apparently, $\boldsymbol{\mathcal{E}} : \mathcal{U} \to \mathbb{R}$ is strictly convex and continuous. Apart from that, for every $v \in \mathcal{U}$, due to Poincaré's inequality applied for each fixed $\boldsymbol{\tau} \in \mathcal{P}$, which is allowed since $v(\boldsymbol{\tau}, \cdot) \in W_0^{1,p}(\Omega(\boldsymbol{\tau}))$ for all $\boldsymbol{\tau} \in \mathcal{P}$, we have that

$$\int_{\mathcal{P}} \int_{\Omega(\boldsymbol{\tau})} |v(\boldsymbol{\tau}, x)|^p \, \mathrm{d}x \, \mathrm{d}\boldsymbol{\tau} \le \int_{\mathcal{P}} \left( 2 \operatorname{diam}(\Omega(\boldsymbol{\tau})) \right)^p \int_{\Omega(\boldsymbol{\tau})} |\nabla_x v(\boldsymbol{\tau}, x)|^p \, \mathrm{d}x \, \mathrm{d}\boldsymbol{\tau} \tag{29}$$
$$\le \left( 1 + 2 \sup_{\boldsymbol{\tau} \in \mathcal{P}} \operatorname{diam}(\Omega(\boldsymbol{\tau})) \right)^p \int_{\mathcal{P}} \int_{\Omega(\boldsymbol{\tau})} |\nabla_x v(\boldsymbol{\tau}, x)|^{p(\boldsymbol{\tau})} \, \mathrm{d}x \, \mathrm{d}\boldsymbol{\tau} \,,$$

which for any $v \in \mathcal{U}$ and $\varepsilon \in (0, 1]$, using for each $\boldsymbol{\tau} \in \mathcal{P}$, the $\varepsilon$-Young inequality with constant $c(p, \varepsilon) \coloneqq \frac{(p\varepsilon)^{1-p}}{p'}$, implies that

$$\boldsymbol{\mathcal{E}}(v) \ge \int_{\mathcal{P}} \frac{1}{p} \int_{\Omega(\boldsymbol{\tau})} |\nabla_x v(\boldsymbol{\tau}, \cdot)|^p \, \mathrm{d}x \, \mathrm{d}\boldsymbol{\tau} - \int_{\mathcal{P}} \int_{\Omega(\boldsymbol{\tau})} c(p, \varepsilon) |f(\boldsymbol{\tau}, \cdot)|^{p'} - \varepsilon |v(\boldsymbol{\tau}, \cdot)|^p \, \mathrm{d}x \, \mathrm{d}\boldsymbol{\tau}$$
$$\ge \left( \frac{1}{p} - \varepsilon \left( 1 + 2 \sup_{\boldsymbol{\tau} \in \mathcal{P}} \operatorname{diam}(\Omega(\boldsymbol{\tau})) \right)^p \right) \int_{\mathcal{P}} \int_{\Omega(\boldsymbol{\tau})} |\nabla_x v(\boldsymbol{\tau}, \cdot)|^p \, \mathrm{d}x \, \mathrm{d}\boldsymbol{\tau} \tag{30}$$
$$- \frac{(p\varepsilon)^{1-p'}}{p'} \int_{\mathcal{P}} \int_{\Omega(\boldsymbol{\tau})} |f(\boldsymbol{\tau}, \cdot)|^{p'} \, \mathrm{d}x \, \mathrm{d}\boldsymbol{\tau} \,.$$

From equation 29 and equation 30, for $\varepsilon > 0$ sufficiently small, using that, by assumption, it holds $\sup_{\boldsymbol{\tau} \in \mathcal{P}} \operatorname{diam}(\Omega(\boldsymbol{\tau})) < \infty$[6], we conclude that from $\|v\|_{\mathcal{U}} \to \infty$, it follows that $\boldsymbol{\mathcal{E}}(v) \to \infty$, i.e., $\boldsymbol{\mathcal{E}} : \mathcal{U} \to \mathbb{R}$ is weakly coercive, so that the direct method in the calculus of variations, cf. Dacorogna (2007), yields the existence of a unique minimizer $u^* \in \mathcal{U}$ of $\boldsymbol{\mathcal{E}} : \mathcal{U} \to \mathbb{R}$.

**ad (ii).** A direct calculation shows that $\boldsymbol{\mathcal{E}} : \mathcal{U} \to \mathbb{R}$ is continuously Fréchet differentiable with

$$\langle D\boldsymbol{\mathcal{E}}(u), v \rangle_{\mathcal{U}} = \int_{\mathcal{P}} \langle DE_{\boldsymbol{\tau}}(u(\boldsymbol{\tau}, \cdot)), v(\boldsymbol{\tau}, \cdot) \rangle_{W_0^{1,p}(\Omega(\boldsymbol{\tau}))} \, \mathrm{d}\boldsymbol{\tau}$$

for all $u, v \in \mathcal{U}$. Therefore, due to the minimality of $u^* \in \mathcal{U}$, for every $v \in \mathcal{U}$, we have that

$$0 = \langle D\boldsymbol{\mathcal{E}}(u^*), v \rangle_{\mathcal{U}} = \int_{\mathcal{P}} \langle DE_{\boldsymbol{\tau}}(u^*(\boldsymbol{\tau}, \cdot)), v(\boldsymbol{\tau}, \cdot) \rangle_{W_0^{1,p}(\Omega(\boldsymbol{\tau}))} \, \mathrm{d}\boldsymbol{\tau} \,. \tag{31}$$

Since $W_0^{1,p}(\Omega(0))$ is separable, there exists a countable dense subset $(\psi_k)_{k \in \mathbb{N}} \subseteq W_0^{1,p}(\Omega(0))$. Also, appealing to (Nägele, 2015, Lemma 2.1), for any $\boldsymbol{\tau} \in \mathcal{P}$, the pull-backs $((\varphi_{\boldsymbol{\tau}}^{-1})^* \psi_k)_{k \in \mathbb{N}} \coloneqq (\psi_k \circ \varphi_{\boldsymbol{\tau}}^{-1})_{k \in \mathbb{N}} \subseteq W_0^{1,p}(\Omega(\boldsymbol{\tau}))$, are dense in $W_0^{1,p}(\Omega(\boldsymbol{\tau}))$. In addition, (Nägele et al., 2017, p. 6 ff.) shows that $(\psi_k)_{k \in \mathbb{N}} \coloneqq (\boldsymbol{\tau} \mapsto (\varphi_{\boldsymbol{\tau}}^{-1})^* \psi_k)_{k \in \mathbb{N}} \subseteq \mathcal{U}$. Next, choosing $v = \varphi \psi_k \in \mathcal{U}$ in equation 31 for arbitrary $\varphi \in C_0^\infty(\mathcal{P})$ and $k \in \mathbb{N}$, we further deduce that

$$\int_{\mathcal{P}} \langle DE_{\boldsymbol{\tau}}(u^*(\boldsymbol{\tau}, \cdot)), \psi_k(\boldsymbol{\tau}, \cdot) \rangle_{W_0^{1,p(\boldsymbol{\tau})}(\Omega)} \, \varphi(\boldsymbol{\tau}) \, \mathrm{d}\boldsymbol{\tau} = 0 \,,$$

so that, owing to the countability of $(\psi_k)_{k \in \mathbb{N}} \subseteq \mathcal{U}$, the fundamental lemma of calculus of variations implies that for a.e. $\boldsymbol{\tau} \in \mathcal{P}$, it holds for all $k \in \mathbb{N}$

$$\langle DE_{\boldsymbol{\tau}}(u^*(\boldsymbol{\tau}, \cdot)), (\varphi_{\boldsymbol{\tau}}^{-1})^* \psi_k \rangle_{W_0^{1,p(\boldsymbol{\tau})}(\Omega)} = 0 \,.$$

---

[6]Here, we exploit that there exists $K > 0$ such that $K^{-1} \le \det(D\varphi_{\boldsymbol{\tau}}) \le K$ in $\Omega(\boldsymbol{\tau})$ for all $\boldsymbol{\tau} \in \mathcal{P}$, cf. (Nägele et al., 2017, (3.1)).

As $((\varphi_{\boldsymbol{\tau}}^{-1})^* \psi_k)_{k \in \mathbb{N}}$ is dense in $W_0^{1,p}(\Omega(\boldsymbol{\tau}))$ for all $\boldsymbol{\tau} \in \mathcal{P}$, we find that for a.e. $\boldsymbol{\tau} \in \mathcal{P}$, it holds for all $v \in W_0^{1,p}(\Omega(\boldsymbol{\tau}))$

$$\langle DE_{\boldsymbol{\tau}}(\boldsymbol{u}^*(\boldsymbol{\tau}, \cdot)), v \rangle_{W_0^{1,p(\tau)}(\Omega)} = 0 \,.$$

Eventually, since for every $\boldsymbol{\tau} \in \mathcal{P}$, the $p$-Dirichlet energy $E_{\boldsymbol{\tau}} : W_0^{1,p}(\Omega(\boldsymbol{\tau})) \to \mathbb{R}$ is strictly convex, for a.e. $\boldsymbol{\tau} \in \mathcal{P}$, the slice $\boldsymbol{u}^*(\boldsymbol{\tau}, \cdot) \in W_0^{1,p}(\Omega(\boldsymbol{\tau}))$ is a unique minimizer of $E_{\boldsymbol{\tau}} : W_0^{1,p}(\Omega(\boldsymbol{\tau})) \to \mathbb{R}$.

**ad (iii) and (iv).** Follow from point (ii) and Theorem 2. $\qquad\square$

