# OpenReview forum: "Breaking the Curse of Dimensionality for Parametric Elliptic PDEs"
_ICLR.cc/2023/Conference — Submitted to ICLR 2023_

### Official Review · Reviewer_jdZY · 2022-10-22

**Confidence:** 5
**Correctness:** 3
**Technical Novelty And Significance:** 2
**Empirical Novelty And Significance:** 1
**Recommendation:** 1

**Clarity, Quality, Novelty And Reproducibility:**

The paper is easy to read, but the main results are incremental. The novelty is not enough for being published in ICLR.

**Strength And Weaknesses:**

Strength: The error estimate in Theorem 1 seems to be new in the study of p-Laplace equation. The key ingredient of the analysis is a Cea type lemma which was proved for the case p=2 previously. The paper extends this to the general case that p\geq 1.

Weakness: The results obtained in the paper are all about the p-Laplacian equation alone and no new results were presented from the side of machine learning (or neural networks).  Hence to me the paper is more suitable for a journal publication in the field of numerical PDEs but not in a conference proceeding of machine learning.

**Summary Of The Paper:**

This paper studied the approximation error of neural networks in approximating a parametric p-Laplace equation with the homogeneous Neumann boundary condition. The main result (theorem 1) established an W^{1,p}-error bound for an approximate solution in terms of the energy difference and the minimum W^{1,p}-error of the underlying approximation function class. The authors also discussed how the error bound can be used to prove that neural networks can beat the curse of dimensionality when the exact solution of the PDE is sufficiently smooth or Barron.

**Summary Of The Review:**

I would not accept the paper in ICLR, but it could be published in a math journal on numerical analysis of PDEs.

---

> ### Author Response · Authors · 2022-11-15
> **Reply to Referee jdZY**
>
> Thank you for your time to read and assess the manuscript.
>
> We understand that the main result of the paper addresses questions mainly related to the structure of the non-linear PDE and does not employ any neural network structure of the ansatz class. However, we would argue that the absence of structure of the ansatz class (it is not a vector space) is indeed special to neural networks.
>
> Furthermore, we believe that the manuscript is still interesting for the audience of a machine learning conference. Compare to the answer of the second review.

---

### Official Review · Reviewer_bRj3 · 2022-10-24

**Confidence:** 3
**Correctness:** 4
**Technical Novelty And Significance:** 1
**Empirical Novelty And Significance:** Not applicable
**Recommendation:** 3

**Clarity, Quality, Novelty And Reproducibility:**


It's clear but not novel.

**Strength And Weaknesses:**

Question 1: What's the application of p-Laplacian equation in high dimension?  ( I don't think clustering is a good application.
Question 2: The main contribution is a Céa's lemma's lemma for PDE. All other part of the proof is not the contribution of the paper, e.g. breaking the curse of dimensionality. It seems this paper is more fittable for a PDE journal but not a machine learning conference. Why this paper will catch ICLRs's audience's interest?
Question 3: The Cea lemma is also not novel. For p-laplacian, it's easy to find a 1989 paper [1] covers it. It can be used for non-linear ansatz for paper [2] is using it for their bound.
Question 4: It would be interesting to see whether this cea lemma will lead to information theoretical optimal bound as [3,4]


[1] Finite element error estimates for non-linear elliptic equations of monotone type

[2] Sheng H, Yang C. PFNN: A penalty-free neural network method for solving a class of second-order boundary-value problems on complex geometries. Journal of Computational Physics, 2021, 428: 110085.

[3] Nickl R, van de Geer S, Wang S. Convergence rates for penalized least squares estimators in PDE constrained regression problems[J]. SIAM/ASA Journal on Uncertainty Quantification, 2020, 8(1): 374-413

[4] Lu Y, Chen H, Lu J, et al. Machine learning for elliptic PDEs: fast rate generalization bound, neural scaling law and minimax optimality[J]. arXiv preprint arXiv:2110.06897, 2021.

**Summary Of The Paper:**

This paper provided a Céa-type lemma's lemma for p-laplacian. Based on this, the author provide a breaking the curse of dimensionality proof for solving the PDE.

**Summary Of The Review:**

The main contribution of the paper may lies in the area of numerical PDE and is not fit for a machine learning conference. At the same time, I don't think the contribution of this paper is significant enough for ICLR.

---

> ### Author Response · Authors · 2022-11-15
> **Reply to Referee bRj3**
>
> Thank you for taking your time to read and assess the manuscript. We will try our best to answer your questions and respond to your concerns.
>
> What is the application of the $p$-Laplacian in high dimensions?
>
> In general, elliptic equations are important in high dimensions, eigenvalue problems in quantum physics are typically eigenvalue problems associated with elliptic operators. In time dependent problems (HJB, Black Scholes) elliptic equations may arise as sub-problems when employing time stepping schemes (minimizing movement for instance).
>
> Beyond the mentioned examples, in engineering applications a PDE is typically not encountered alone, but in context of a physical system. Often, these are multiphysics problems with many parameters (such as material properties, parametrized geometries,...). From this perspective any PDE is interesting in a high dimensional context, namly for exploring the behavior when varying the (physical) parameter space. In fact, we believe that this application alone is sufficient motivation to consider high dimensional problems involving the the $p$-Laplacian.
>
> Why will this paper catch ICLRs's audience's interest?
>
> We believe that people interested in NN based approaches to solving PDEs can benefit from our contribution and that it is accessible also to non-experts in the field, compare to the first review. It is a modular result that is easy to apply and fits well with different possible applications - it can be combined with approximation theorems for classical functions spaces, i.e., Sobolev spaces, but also with more modern ansatz classes such as Barron spaces. Naturally, the methods to establish the result are from the numerical analysis community, but this is not main focus in the current version of the manuscript.
>
> The Cea lemma is also not novel.
>
> For p-laplacian, it's easy to find in a 1989 paper, see [1] and for nonlinear ansatz classes [2]} We inspected both references and are not fully convinced. In [2], they state the results of [1] as Lemma 1. Tracing back to [1] they appear to be using Theorem 1 on page 386 in [1]. Inspecting the proof thereof in [1] the first inequality which we reproduce now in the terms of [1] is as follows
> \begin{equation*}
>     \langle Au - Au_h, u - u_h \rangle + \langle Gu - Gu_h, u - u_h \rangle = \langle Au - Au_h, u-v_h \rangle + \langle Gu - Gu_h, u - v_h \rangle
> \end{equation*}
> In this notation $u$ is the continuous solution, $u_h$ is the FEM solution in a finite dimensional subspace and $v_h$ is an arbitrary function in the finite dimensional subspace. Note that this equality is nothing but Galerkin orthogonality! We therefore strongly believe that one cannot argue in this way.
>
> However, there are ideas in section 7 of [1] that eventually might lead to a similar result to our contribution. However they are neither obvious nor employed in [2]. We therefore believe that our contribution is valuable and novel. Finally, note also that we carefully track all constants for dependency on the dimension of $\Omega$ \& allow for inexact optimization (by introducing $\delta$ in the formulation of our C\'ea Lemma).
>
> It would be interesting to see whether this Cea Lemma will lead to information theoretical optimal bound as [3,4]
>
> We agree that this is an interesting question, albeit beyond the scope of our manuscript.

---

> > ### Comment · Reviewer_bRj3 · 2022-11-15
> > **Be specfic?**
> >
> > Elliptic equations are important in high dimensions doesn’t means p-laplacian equation is important in high dimension.
> >
> >
> > Often, these are multiphysics problems with many parameters (such as material properties, parametrized geometries,...).   For this case, I think most of the PDE is static schrodinger Equation. Can author specific point to the paper using p-laplacian in high dimension to predict the material properties.
> >
> >
> > I still can’t distinguish the difference, but seems potential interest. Can the author be specific for this statement.
> >
> >
> >  it is accessible also to non-experts in the field doesn’t mean it’ll catch interest. The first review also claimed that “As a matter of fact, I am not entirely sure that ICLR is a perfect venue for this type of technical work.” We can’t say we present a cea type lemma s NN solving pde, then it’ll catch  iclr’s audience attention. The main contribution part have no relationship with NN but all about p-laplacian.

---

> > > ### Author Response · Authors · 2022-11-16
> > > **More Details**
> > >
> > > "Can author specific point to the paper using p-laplacian in high dimension to predict the material properties. I still can’t distinguish the difference, but seems potential interest. Can the author be specific for this statement."
> > >
> > > I guess there is some misunderstanding here. We agree that $p$-laplacians with $p\neq 2$ are typically encountered in one to three spatial dimensions. However, parametric right-hand sides, for instance, lead to a high dimensional problem. We want to solve the equation for many different right-hand sides and view these right-hand sides as parametrized, the way we want to solve this is to make every parameter an additional input to the neural network. Note that this idea is not new and even standard practice in the PINNs community, we refer to the documentation of Nvidia modulus, see [a] for the attached link. Once a model is trained over a (physical) parameter space it can be used as a surrogate model for fast inference time for (physical) parameter instance or for optimization over the parameter space.
> > >
> > > For a concrete example of the above involving p-Laplace (like) equations consider flow of electrorheological fluids. In electrorheological fluids the stress-strain relationship of the velocity has a p-structure, includes material properties properties and the electrical field variable and finally the electrical field enters the right-hand side of the equation for the velocity. For the concrete equations compare to [b], for instance equations (0.1)-(0.3) on page viii and ix.
> > >
> > > Other interesting parametric dependencies can stem from moving or parametrized domains, we refer again to the example in [a], for moving domains in the context of the p-Laplacian see [c]. As another example for p-Laplace equations consider thermo-rheological viscous flows see [d].
> > >
> > >
> > > "Elliptic equations are important in high dimensions doesn’t means p-laplacian equation is important in high dimension."
> > >
> > > The case $p=2$ is definitely important for a high dimensional domain. The case $p\neq 2$ less so for a high-dimensional domain, but for parametric problems.
> > >
> > > "The main contribution part have no relationship with NN but all about p-laplacian."
> > >
> > > We saw the contribution fit as it tailors PDE analysis to the need of NN based methods for solving PDEs, in that sense the lack of structure of the ansatz class is what makes it specific to NNs. We believe the result is useful to know and to build up on.
> > >
> > >
> > > References:
> > >
> > > [a] https://docs.nvidia.com/deeplearning/modulus/user_guide/advanced/parametrized_simulations.html
> > >
> > > [b] Electrorheological FLuids: Modeling and Mathematical Theory, Michael Ruzicka, Lecture Note in Mathematics
> > >
> > > [c] Generealized Newtonian fluids in moving domains, Philipp Naegele and Michael Ruzicka, Journal of Differential Equations
> > >
> > > [d] On stationary thermo-rheological viscous flows, Stanislav Antontsev and Jose Rodrigues, Annali dell'Universita di Ferrara

---

> > > > ### Comment · Reviewer_bRj3 · 2022-11-16
> > > > **still have concern**
> > > >
> > > > electrorheological flow and thermo rheological flow are all three dimension equations, do the author considers three dimension are high dimension?
> > > >
> > > > At the same time, I still can’t distinguish the difference between this work and the reference on cea type Lemma in the reference. Can the author help to be specific?

---

> > > > > ### Author Response · Authors · 2022-11-17
> > > > > **More explanation**
> > > > >
> > > > > "electrorheological flow and thermo rheological flow are all three dimension equations, do the author considers three dimension are high dimension?"
> > > > >
> > > > > No of course not. A toy example of what we mean is the following: Solve $$-\Delta_p u = \alpha_1 f_1 + \alpha_2 f_2 + \dots + \alpha_n f_n$$
> > > > >
> > > > > To solve it all together make a net $NN$ that takes inputs $NN(\alpha_1,\dots,\alpha_n, x_1, x_2, x_3)$. Then the net $NN$ must approximate a multivariate function of many variables, hence a high dimensional object.
> > > > > We stress again that this approach is common practice, see again reference [a]. The references above give examples of non-toy problems for these types of equations, where the parameters $\alpha_1,\dots,\alpha_n$ correspond to physical quantities. Does this make it clearer what we mean?
> > > > >
> > > > > "At the same time, I still can’t distinguish the difference between this work and the reference on cea type Lemma in the reference. Can the author help to be specific?"
> > > > >
> > > > > Sure. What is it that we should expand on? Generally the problem is the following: the stated reference works for linear ansatz classes. Then you know that on a subspace the problem is solved for a restricted class of test functions. This leads to Galerkin orthogonality relations. Note that this in its very core depends on the ansatz space being linear.

---

### Official Review · Reviewer_iQb8 · 2022-10-24

**Confidence:** 1
**Correctness:** 4
**Technical Novelty And Significance:** 4
**Empirical Novelty And Significance:** Not applicable
**Recommendation:** 10

**Clarity, Quality, Novelty And Reproducibility:**

I can only comment on the first 4 pages that I have found very well written for the non-expert, with enough references to the current literature and high-level description of the consequences of Theorem 1.

**Strength And Weaknesses:**

The text is clearly written, with a well written introductory section and a very useful section following the statement of Theorem 1 discussing the consequences of the main results and how it fits within the modern literature. As a non-expert, I have been to easily follow this sections and understand the importance of the stated result.

Now, I need to admit right away that I am not an expert in this area and have not been able to follow the proof of the main result. As a matter of fact, I am not entirely sure that ICLR is a perfect venue for this type of technical work. Most of the details are in the appendix and I am not entirely sure that the ICLR reviewing process is adapted for proof-checking this type of (important) contributions.

I am sorry that I cannot provide useful comments to the authors (and to the AC).

**Summary Of The Paper:**

The text focuses on estimating solution of the PDE $\text{div}[|\nabla u|^{p-2} \nabla u ] = f$. The solution $u_\star$ to this PDE can be expressed as minimiser of an energy functional, $u_\star = \text{argmin}  \mathcal{E}(u)$. The main result of the paper, Theorem 1, shows that any function $v$ in a (very general) subset of function $\widetilde{M}$ is such that $|\nabla v - \nabla u_\star|_{L^p}$ can be upper bounded by a constant multiple of the sum of the energy gap $A = \mathcal{E}(v) - \text{inf}\big(  \mathcal{E}(v') : v' \in \widetilde{M} \Big)$ and the quantity $B = \text{inf}\Big( |\nabla v' - \nabla u_\star| : v' \in \widetilde{M} \Big)$ related to the quality of the approximating subspace $\widetilde{M}$.

Since the quantity $B$ is quite well studied (i.e. approximation theory of neural network spaces), this shows that any algorithm that can minimise the energy functional leads to good approximation to the solution $u_\star$.

**Summary Of The Review:**

1. **I am NOT knowledgeable enough to comment on the quality / novelty / correctness of the article**
2. I am inclined to think that ICLR is not the right venue for this type of work that requires serious proof-checking.

---

> ### Author Response · Authors · 2022-11-15
> **Reply to Referee iQb8**
>
> Thank you for your feedback. We are happy to hear that even for the non-expert in the field the high-level description of the manuscript was understandable.

---

### Decision · Program_Chairs · 2023-01-20

**Decision:**

Reject

**Justification For Why Not Higher Score:**

Conceptually, it's not clear the paper teaches us very much about when and why neural networks specifically (as opposed to other universal approximators) would be a good class of functions to use for PDE solvers. Technically, the approach is fairly standard apart from a new Cea type lemma which is unlikely to be of interest to (or appreciated by) the broader ICLR community.

**Justification For Why Not Lower Score:**

N/A

**Metareview: Summary, Strengths And Weaknesses:**

The paper concerns two representational results on the solutions of PDEs using neural networks: one based on showing the solution has sufficient "regularity" (plus using standard non-parametric estimation rates), the other (minor one) assuming the solution has a small Barron norm. Overall, the reviewers thought the paper doesn't have enough content of interest for the ICLR community. For the former (major) result, the main new ingredient is a Cea type lemma which as a standalone result is likely not of interest to the broader machine learning community. The latter result is standard --- and crucially, the authors do not provide conditions under which the solution has small Barron norm (just indicate that *if the Barron norm is small*, the solution can be represented using a small neural network --- which is a standard result). Conceptually, it's also not clear what these results teach us about neural networks specifically (as opposed to any other type of universal approximator that benefits from smoothness). It should be noted that the high-score reviewer (iQb8) explicitly noted during the openreview discussion that they have insufficient expertise to have a strong and informed opinion on the paper, and their score "10" was simply indicative of this fact.